



# A meteorological and chemical overview of the DACCIWA field campaign in West Africa in June–July 2016

Peter Knippertz[1], Andreas H. Fink[1], Adrien Deroubaix[2], Eleanor Morris[3], Flore Tocquer[4], Mat J. Evans[3], Cyrille Flamant[5], Marco Gaetani[5], Christophe Lavaysse[6], Celine Mari[4], John H. Marsham[7], Rémi Meynadier[8], Abalo Affo-Dogo[9], Titike Bahaga[1], Fabien Brosse[4], Konrad Deetz[1], Ridha Guebsi[5], Issaou Latifou[9], Marlon Maranan[1], Philip D. Rosenberg[7], and Andreas Schlueter[1]

[1]Institute of Meteorology and Climate Research, Karlsruhe Institute of Technology, 76128 Karlsruhe, Germany
[2]Laboratoire de Météorologie Dynamique, Ecole Polytechnique, IPSL Research University, Ecole Normale Supérieure, Université Paris-Saclay, Sorbonne Universités, UPMC Univ Paris 06, CNRS, 91128 Palaiseau, France
[3]Wolfson Atmospheric Chemistry Laboratories, Department of Chemistry, University of York, York, YO10 5DD, UK
[4]Laboratoire d'Aérologie, Université de Toulouse, CNRS, UPS, 31400 Toulouse, France
[5]LATMOS/IPSL, Sorbonne Universités, UPMC Univ Paris 06, UVSQ, CNRS, 75252 Paris, France
[6]European Commission, Joint Research Centre, Ispra (VA), Italy
[7]School of Earth & Environment/National Centre for Atmospheric Science, University of Leeds, Leeds LS2 9JT, UK
[8]AXA Group Risk Management Department, Paris, France
[9]Direction Générale Météo Nationale, B.P. 1505, Lomé, Togo

*Correspondence to*: Peter Knippertz (peter.knippertz@kit.edu)

**Abstract.** In June and July 2016 the Dynamics-Aerosol-Chemistry-Cloud Interactions in West Africa (DACCIWA) project organised a major international field campaign in southern West Africa (SWA) including measurements from three inland ground supersites, urban sites in Cotonou and Abidjan, radiosondes and three research aircraft. A significant range of different weather situations was encountered during this period, including the monsoon onset. The purpose of this paper is to characterise the large-scale setting for the campaign as well as synoptic and mesoscale weather systems affecting the study region in the light of existing conceptual ideas, mainly using objective and subjective identification algorithms based on (re-) analysis and satellite products. In addition, it is shown how the described synoptic variations influence the atmospheric composition over SWA through advection of mineral-dust, biomass-burning and urban-pollution plumes.

The boreal summer of 2016 was characterised by Pacific La Niña, Atlantic El Niño and warm eastern Mediterranean conditions, whose competing influences on precipitation led to an overall average rainy season. During the relatively dusty pre-onset Phase 1 (1–21 June 2016), three westward propagating coherent cyclonic vortices between 4 and 13°N modulated winds and rainfall in the Guinea coastal area. The monsoon onset occurred in connection with a marked extratropical trough and cold surge over northern Africa, leading to a breakdown of the Saharan heat low and African easterly jet and a suppression of rainfall. During this period, quasi-stationary low-level vortices associated with the trough transformed into more tropical, propagating disturbances resembling an African easterly wave (AEW). To the east of this system, moist southerlies penetrated deep into the continent. The post-onset Phase 2 (22 June–20 July 2016) was characterised by a significant increase of low-level cloudiness, unusually dry conditions and strong northeastward dispersion of urban pollution





plumes in SWA as well as rainfall modulation by westward propagating AEWs in the Sahel. Around 12–14 July 2016 an interesting and so-far undocumented cyclonic-anticyclonic vortex couplet crossed SWA. The anticyclonic centre had its origin in the southern hemisphere and transported unusually dry air filled with aged aerosol into the region. During Phase 3 (21–26 July 2016), a similar vortex couplet slightly farther north created enhanced westerly moisture transports into SWA

and extraordinarily wet conditions, accompanied by a deep penetration of the biomass-burning plume from central Africa. Finally, a return to more undisturbed monsoon conditions took place during Phase 4 (27–31 July 2016). The in-depth synoptic analysis reveals that several significant weather systems during the DACCIWA campaign cannot be attributed unequivocally to any of the tropical waves and disturbances described in the literature, and thus deserve further study.

## 1 Introduction

The atmosphere over summertime West Africa is influenced by processes covering a wide range of scales, which can interact with each other in complex ways (Lafore et al., 2010; Redelsperger et al., 2006). The dominating phenomenon is the West African monsoon (WAM), which is mainly driven by the surface pressure contrast between the relatively cool waters of the eastern tropical Atlantic Ocean and the Saharan heat low (SHL). The former is related to the installation of the Atlantic Cold Tongue (ACT) starting in April-May and reaching its maximum horizontal extension in mid-August (Caniaux

et al., 2011). At the equator, colder SSTs increase the stability of the marine atmospheric boundary layer and decreases the vertical mixing of momentum, leading to weaker surface southerlies (Wallace et al., 1989), while north of the equator as far as the Guinean Coast, the large meridional SST gradient strengthens the surface wind through a hydrostatically-induced meridional pressure gradient (Lindzen and Nigam, 1987). This creates a low-level circulation characterised by surface wind divergence and subsidence over the equator and convergence and convection close to the Guinean Coast in the period before

the full onset of the WAM.

The SHL is a lower tropospheric thermal depression in the Sahara desert west of 10°E, which develops in response to the intense surface heating during boreal summer (Lavaysse et al., 2009). The monsoon typically sets in quite abruptly around the end of June accompanied by a shift of the area of main rainfall from the Guinea Coast to the Sahel (Fitzpatrick et al., 2015; Sultan and Janicot, 2003). This event is usually preceded by a northward shift of the so-called Intertropical

Discontinuity (ITD), the near surface confluence zone between southwesterly and northeasterly winds, which marks a northern limit of rainfall occurrence (Fitzpatrick et al., 2016; Lélé and Lamb, 2010). After the monsoon onset, the mid-tropospheric circulation over West Africa is dominated by the African easterly jet (AEJ), which is caused by the strong meridional temperature and moisture gradient at low levels (Cook, 1999; Wu et al., 2009). The AEJ is maintained by the anticyclonic circulation associated with the monsoonal subsidence, which characterises the mid-upper troposphere over the

Sahara (Chen, 2005; Thorncroft and Blackburn, 1999) and above the shallow dry convection in the SHL (Garcia-Carreras et al., 2015; Ryder et al., 2015). Barotropic and baroclinic instabilities associated with the jet create an environment favourable to the generation of African easterly waves (AEWs) (Thorncroft and Hoskins, 1994a, 1994b; Wu et al., 2012), synoptic-scale





disturbances characterised by a 2–6-day period in the Sahel.

On multidecadal to interannual timescales, WAM variability is strongly associated with global sea surface temperature (SST) anomalies (Rodríguez-Fonseca et al., 2015; Rowell, 2013). For example, positive phases of the Atlantic multidecadal variability favour precipitation in the Sahel (Ting et al., 2011; Zhang and Delworth, 2006). On interannual timescales, SST

variability in the tropical Atlantic modulates the land-sea thermal gradient, leading to meridional displacements of the precipitation belt over West Africa (Losada et al., 2010; Polo et al., 2008). SSTs over the Mediterranean Sea influence the amount of moisture being transported across the Sahara desert and converging over the eastern Sahel (Fontaine et al., 2010; Gaetani et al., 2010). Interannual variability of the WAM is also influenced by the SST variability in the tropical Indian/Pacific Oceans, which may trigger stationary waves along the equator interacting over the Sahel (Mohino et al., 2011;

Rowell, 2001).

On intraseasonal to synoptic timescales, an important source of variability is the propagation of convectively coupled equatorial wave (CCEW) disturbances and the Madden-Julian Oscillation (MJO) (Mohino et al., 2012; Pohl et al., 2009). In addition, variability of the SHL strength and position modulate the distribution of the monsoonal precipitation in the Sahel in the zonal direction. During periods of a deeper SHL, the shallow cyclonic circulation associated with the thermal low is

intensified, strengthening the Atlantic westerly flow and the convergence in the Sahel, which leads to wet (dry) anomalies in the eastern (western) Sahel (Lavaysse et al., 2010b). The SHL phases are modulated at synoptic timescales by both tropical and mid-latitudes disturbances (Chauvin et al., 2010; Lavaysse et al., 2010a). Variations in the intensity and position of the AEJ influence the location, amplitude and propagation speed of AEWs, which play a crucial role in the modulation of convective precipitation in the Sahel, mainly through their influence on thermodynamics and vertical wind shear (Gu et al.,

2004; Skinner and Diffenbaugh, 2013). To the south of the Sahel, low-level vortices unrelated to AEWs can affect rainfall (e.g. Fink et al., 2006). Convection in West Africa is often organised on the mesoscale, particularly in the form of fast propagating squall line systems (Fink and Reiner, 2003), but more isolated thunderstorms or showers also occur, e.g. triggered by the sea-breeze convergence along the Guinea Coast (Fink et al., 2010). The moist convection embedded within the monsoon flow has been shown to be intrinsic to the monsoon, and the poor representation of convection in models leads

to biases in the WAM (Birch et al., 2014; Garcia-Carreras et al., 2013; Marsham et al., 2013).

The atmospheric composition over southern West Africa (SWA hereafter) during the wet season is a complex combination of air masses transported from remote sources bringing desert dust or biomass burning aerosol, and local anthropogenic pollution (Mari et al., 2011). The Sahara desert to the north of the region is the largest aerosol source in the world and the transport of dust southwards is a significant source of aerosol for SWA (e.g. Chiapello, 2014; Shao et al., 2011). Forest

fires in the immediate region are not thought to be significant in this period, but the transport of biomass burning species from the southern hemisphere (SH) has been observed (Mari et al., 2008). Anthropogenic emissions from the combustion of fossil fuels, biofuels and refuse are on the rise and expected to keep increasing significantly in the near future due to the rapid growth of cities in the region (Knippertz et al., 2015b; Liousse et al., 2014). Air quality thus is a concern with multiple sources of anthropogenic emissions from domestic open fires, road traffic, street dust, waste burning, oil extraction and





refining, ships, industrial activity, power plants etc. SWA is also characterized by a south-north gradient of vegetation from rain forest in the coastal belt to the sub-Sahelian savannah in the north. The dense vegetation can emit large quantities of biogenic compounds (isoprene, monoterpenes etc.), which profoundly alter the gas and aerosol composition in the region (Mari et al., 2011). The relative role of local biogenic and anthropogenic emission, the long-range transport of other

compounds into the SWA atmosphere, coupled to the peculiar dynamics of the region during the monsoonal period leads to a chemically complex region.

A lack of an observational network adequate to better understand processes and to evaluate model simulations and satellite data has for a long time impeded scientific progress in West Africa and motivated the organisation of large international field campaigns. An early example, which revolutionized the understanding of the WAM system at that time, is the GARP

Atlantic Tropical Experiment (GATE) (Kuettner, 1974). The largest such programme in recent decades is the African Monsoon Multidisciplinary Analysis (AMMA), which took place in 2006 with a focus on Sahelian convection (Lebel et al., 2010). More recently, the Dynamics-Aerosol-Chemistry-Cloud Interactions in West Africa (DACCIWA) project (Knippertz et al., 2015a) organised a major international field campaign during June and July 2016, focusing for the first time on the most populated southern coastal region of West Africa. In addition to a number of meteorological aspects, the DACCIWA

campaign also had a focus on atmospheric composition, including questions of air pollution and cloud-aerosol interactions (Knippertz et al., 2015b). Field activities included measurements from three inland ground supersites (Savé in Benin, Kumasi in Ghana, Ile-Ife in Nigeria), urban sites (Cotonou in Benin, Abidjan in Ivory Coast), radiosondes and three research aircraft stationed in Lomé (Togo). A detailed description of the field activities is given in Flamant et al. (2017, submitted).

The objectives of this paper are to (a) place the campaign period June-July 2016 into a larger-scale climatological context,

(b) describe the behaviour of the WAM system (e.g. onset, AEJ and SHL positions), (c) characterise the most important synoptic-scale weather systems affecting SWA (e.g. AEWs, vortices), and (d) discuss impacts on rainfall, clouds and atmospheric composition. This way the paper aims to fulfil a similar role as Janicot et al. (2008) for AMMA. The analysis will build on and expand some of the concepts introduced in this section, and provide a consistent framework for the detailed analysis of DACCIWA field campaign data in following years. From an atmospheric dynamics and chemistry perspective

SWA is of particular interest and has not been studied much in the past. AMMA had fewer stations in SWA and only few publications covering this region. During GATE, data quality (e.g. from radiosondes and satellites) and data assimilation (e.g. use of cloud motion vectors) had not evolved enough to allow a reliable analysis of 850-hPa streamlines for example, especially over the Gulf of Guinea (e.g. Sadler and Oda, 1979). Relying on the densest radiosonde network at the Guinea Coast and in the Soudanian zone ever, DACCIWA can for the first time provide a detailed account of SWA weather systems

and their impacts on precipitation and atmospheric composition. In contrast to the Sahel, SWA is often characterised by situations with high moisture and relatively low convective inhibition (CIN), while the vertical wind shear is typically weak. Thus convection is relatively easy to trigger and remains less-well organised, yet brings substantial rains. These are often connected to weak and vertically shallow cyclonic and anticyclonic vortices (e.g. Fink et al., 2006), but details of this relationship are still unclear as are their linkage to classical equatorial-wave / AEW disturbances. This paper will shed some



new light into these fundamental unexplored dynamical features including the specific question of onset dynamics.

The paper is structured as follows: In sect. 2 an overview of the employed data and methods will be given. Section 3 contains a relatively short discussion of the large-scale settings followed by a more detailed analysis of the synoptic-scale evolution in sect. 4. Section 5 discusses the implication of meteorological variation on atmospheric composition, focusing on

Saharan/Sahelian dust, biomass-burning aerosol from the SH and pollution plumes from the cities along the Guinea Coast. Main conclusions will then be given in sect. 6.

## 2 Data and methods

### 2.1 Data

For the investigation of atmospheric dynamics, analysis and re-analysis data from the European Centre for Medium-Range

Weather Forecasts (ECMWF) are used. Most of the analysis are based on the ERA-Interim (hereafter ERA-I) re-analysis at about 0.7° grid spacing (Dee et al., 2011), which allow the computation of background climatologies back to 1979. For investigations focusing on the campaign period in 2016 alone the higher-resolution operational analyses (native resolution of ~9km; model version Cy41r2, see www.ecmwf.int) are employed. As there was no change to the operational system during the study period, these data can be regarded as homogeneous, in contrast to longer timespans of operational data. The

majority of radiosondes launched during the DACCIWA field campaign were distributed through the Global Telecommunication System and were assimilated at ECMWF. For the analysis of ocean influences on West Africa, the 0.25° daily Optimum Interpolation Reynolds SST data are used. The dataset combines observations from different platforms (satellites, ships, buoys) on a regular global grid. A spatially complete SST map is produced by interpolating to fill in gaps (Reynolds et al., 2007). Data have been retrieved from the NOAA NCDC (National Oceanic and Atmospheric

Administration - National Climatic Data Center) ftp site (http://www.ncdc.noaa.gov). Monthly anomalies for June-July 2016 and daily anomalies are based on the 1981–2016 climatology.

As a precipitation estimate, the standard Tropical Rainfall Measuring Mission (TRMM) product 3B42 (V7) with 0.25° grid spacing is used (Huffman et al., 2007). This product combines information from space-borne radar, microwave and infrared channels, subject to monthly calibration with surface rain gauges if available. Since September 2014, the real-time

calibration of microwave radiances using the precipitation radar ceased due to the decommissioning of the TRMM satellite and was replaced by using climatologically adjustments. Although this caused a discontinuity, the TRMM 3B42 product was prioritised over the Global Precipitation Measurement (GPM) Integrated Multi-satellitE Retrievals for GPM (IMERG) successor product due to the longer availability (1998–2016), which allowed for calculation of anomalies. The temporal resolution of this product is 3 hourly, but here daily accumulations are used for most investigations. In addition, outgoing

longwave radiation (OLR) data from the Spinning Enhanced Visible and Infrared Imager (SEVIRI) on the geostationary Meteosat Second Generation (MSG) satellites with a spatial and temporal resolution of ~3 km at nadir and 15 minutes, respectively, are used as a proxy for convective activity (Schmetz et al., 2002). In particular, channel 9 (i.e. approximately





9.80–11.80 μm) of the thermal infrared band is taken to retrieve cloud-top temperatures and to ensure day and night coverage. Different types of clouds are analysed using information of cloud top characteristics (CTX) and the cloud mask (CMA) from the Satellite Application Facility on Climate Monitoring (CM SAF). Both the CTX and CMA subsets are part of the CLAAS-2 (CM SAF Cloud Property Dataset Using SEVIRI, V2) dataset, which is derived from information provided

by SEVIRI (Stengel et al., 2013). Therefore, CLAAS data have the same temporal and spatial resolution as the SEVIRI dataset.

As this paper is meant to give a broad overview of meteorological and chemical conditions only, a detailed analysis of DACCIWA field campaign data is left to follow-up studies. The only exception is radiosonde data from Abidjan (for location, see Fig. 1) used to illustrate a period of unusual dryness during July 2016. Relative humidity was derived from

four-times daily (00, 06, 12 and 18 UTC) soundings using the high-resolution vertical profiles obtained from the MODEM radiosonde system. The analysis will concentrate on the main DACCIWA study region (8°W–8°E, 5–10°N, see Fig. 1), but influences on that region from a much wider area will be considered.

### 2.2 Methods

In order to better understand and characterise atmospheric variability during the DACCIWA campaign, a number of features

important for SWA were objectively or subjectively identified:

1) Equatorial waves: The presence of CCEWs is identified using the wave filtering method in specific wavenumber-frequency domains as described in Wheeler and Kiladis (1999). Additional to the CCEWs (Kelvin waves, Madden-Julian Oscillation, mixed Rossby-gravity and equatorial Rossby waves), tropical depression-like disturbances (TD) are filtered following the method by Roundy and Frank (2004). These often correspond to AEWs over West Africa. The filtering is

applied to the 3-hourly TRMM 3B42 (V7) precipitation dataset (see sect. 2.1) within the northern equatorial band 5°–15°N, which contains the bulk of the convective precipitation during the campaign period but excludes some heavier oceanic rainfalls (see Fig. 5b).

2) Heat low index: Following Lavaysse et al. (2009), the low-level atmospheric thickness between 925 and 700hPa over a domain that covers West and North Africa is used to determine the location and the intensity of the SHL. The location

corresponds to the region with thickness values larger than the 90th percentile. The intensity is defined directly through the thickness in gpm, indicating the thermal dilation of the lower atmosphere. Once the SHL is detected and the intensity of each grid point calculated, the centre of the SHL is defined as the barycentre in longitude and latitude. These computations are based on ERA-I (see sect. 2.1).

3) AEJ index: Average position and strength of the AEJ are objectively calculated based on Berry et al. (2007). Six-hourly

ERA-I winds at 700hPa within the region 0–30$^0$N and 8$^0$W–8$^0$E (longitudinal extent of DACCIWA focus region, see Fig. 1) are used. A spatial low-pass filter with a cutoff wavelength of 1000 km is applied to calculate shear vorticity, which is then used to determine the jet axis. The average wind speed along the jet axis and the mean latitudinal position is estimated for June-July 2016 and the long-term climatology (1987–2016) for comparison.





4) MCS identification: The evolution of deep convective clouds are monitored by applying an overlap-based tracking algorithm (Mathon and Laurent, 2001; Schröder et al., 2009; Williams and Houze, 1987) to the 15-minute infrared data of SEVIRI. In two successive images, cold cloud regions are identified first and then connected in time by determining the highest accordance with respect to area, area overlap and spatial translation. Here, deep convective clouds are defined as regions with a brightness temperature of $\leq 233$ K and an area of least 100 contiguous pixels (i.e. ~900 km²). The former criterion is widely used as a proxy for deep precipitating convection in tropical regions, whereas the latter excludes convective systems with low contribution to total cold cloud cover (Mathon and Laurent, 2001; Schröder et al., 2009).

5) Synoptic-scale vortices: Close inspection of daily weather charts suggests that only few classical AEWs occurred during the study period and that a more flexible approach is needed to fully represent the observed richness of coherent features. After some testing, a combination of subjective tracking of vortex centres from unfiltered 850-hPa streamlines with Hovmöller plots of 850-hPa vorticity and meridional wind was selected.

6) Long-range transport of biomass burning and dust enriched air masses: Biomass burning plumes from central Africa transported into the domain were tracked with carbon monoxide (CO) calculated from the ECMWF Copernicus Atmosphere Monitoring Service-Integrated Forecasting System (CAMS-IFS; Inness et al., 2013). Dust plumes from Sahelian and Saharan sources, north of the DACCIWA domain, were identified using the CAMS-IFS dust aerosol optical depth (DAOD). Where available CAMS-IFS assimilates satellite information to bring the model output closer to reality.

7) Turbulent dispersion of urban plumes from the five major cities, where DACCIWA aircraft and ground measurements were taken (Abidjan, Kumasi, Accra, Lomé, Cotonou, see Fig. 1), were calculated daily using forward trajectories of passive tracers in a Lagrangian framework. Two models were used: (a) FLEXPART v6.2 (https://www.flexpart.eu/; Stohl et al., 2005) driven with ECMWF ERA-I winds and (b) HYSPLIT v4.8 (Stein et al., 2015) driven by GDAS (Global Data Assimilation System) winds. Both models were run for 24 hours with a continuous emissions of the tracer. The extent of the plume was calculated in FLEXPART using the root mean square distance of particles from the source at the end of the 24 hour simulation. For HYSPLIT the plume boundary was defined at the end of the 24 hours at a threshold concentration of 10–14 unit m$^{-3}$, where one unit of tracer was emitted from the source in 24h.

### 3 Large-scale settings

This short section aims to characterise the large-scale setting the DACCIWA field campaign period, June-July 2016, was embedded in. Figure 2 shows global SST anomalies for June-July 2016. While at the beginning of the year El Niño conditions were still prevalent (not shown), by June a transition to La Niña had occurred, which usually favours monsoonal precipitation in the Sahel (Joly and Voldoire, 2009). At the same time, the equatorial Atlantic Ocean was relatively warm with widespread anomalies above 1 K (Fig. 2). These warm events, sometimes referred to as Atlantic El Niños (Okumura et





al. 2006), are associated with a suppressed ACT and are linked with westerly surface wind perturbations at the equator. The reduced surface wind stress causes less surface oceanic divergence and vertical mixing, leading to reduced SST cooling. This reduces the pressure gradient towards the SHL and thus the inland penetration of monsoonal rains. Since the 1970s, a frequent anticorrelation between the Atlantic and Pacific El Niño's has been observed (Rodríguez-Fonseca et al., 2015).

Warmer equatorial waters in the Gulf of Guinea as in 2016 exhibit a strong correlation with above-normal rainfall at the Guinea Coast, which has been robust throughout the 20[th] century (Diatta and Fink, 2014). Mohino et al. (2011) argue that a warm eastern equatorial Atlantic Ocean and a simultaneous cold eastern Pacific Ocean exert compensational forcings on Sahelian rainfall, such that the archetypical dipole response during warm years in the Gulf of Guinea has rarely been observed after the 1970s. In the Mediterranean Sea, positive SST anomalies are found over the eastern basin accompanied by

negative anomalies in the northwestern part of the Indian Ocean (Fig. 2). Positive SST differences between these two areas are associated with rainfall excess over the Sahel (Fontaine et al., 2011; Park et al., 2016). Overall, it appears that a compensation of these different factors has been in place in 2016, since the June – September Sahelian rainfall was only very slightly above normal (not shown).

For the DACCIWA focus region (Fig. 1) and along most of the Guinea Coast June–September rainfall turned out to be

normal (not shown), despite the extended dry spell during Phase 2 discussed in sect. 4 (Fig. S1b). Only the Guinea and Cameroon Line Mountains and the Gulf of Bonny had above-normal rainfall. It is unclear why the warmer waters in the Gulf of Guinea did not cause more rainfall in lowland areas. A possible explanation are warm SSTs in the South Atlantic (Fig. 2), which seem to reduce the positive effect of the equatorial Atlantic (Nnamchi and Li, 2011). In 2016, the situation was further complicated by relatively cold SSTs along the coasts of Senegal and Ghana-Togo, whose impacts on rainfall are not clear.

An inspection of the daily evolution of zonally averaged (10°W–4°E) SSTs over the tropical eastern Atlantic during June-July 2016 reveals the typical establishment of the equatorial cold tongue (1°N–5°S) and of upwelling of cooler water along the Guinea Coast (4–6°N) (Fig. 3). The onset of the ACT occurred around 10 June 2016 (mean date is June 11 with a standard deviation of 12 days according to Caniaux et al. (2011)) with SSTs slightly below the long-term average, followed by a significant warming and southward retreat between mid-June and 05 July 2016 with warm anomalies surpassing 1.5 K.

After that, the ACT gets re-established but absolute temperatures stay above average by 0.5 K or more until the end of July 2016, consistent with the anomalies shown in Fig. 2. The coastal upwelling sets in much later (Fig. 3). First indications of a cooling are found around 18 June 2016, but a more substantial cooling begins on 27 June 2016 until SSTs drop below 26°C across a broader coastal strip until the end of July. The phenomenon is a little stronger in 2016 than in other years with negative anomalies on the order of 0.5 K, particularly along 5°N.

The other important driver of the WAM is the SHL. Figure 4 shows its intensity and position on a daily basis during June-July 2016. During the first three weeks of June, the SHL is in an intense phase (Fig. 4a) and shows large east–west fluctuations with a period of about 10 days, remaining mostly to the east of the climatological position (Fig. 4b). The SHL is also located further to the north than usual, associated with large positive temperature anomalies over northeastern Africa and anomalous southwesterly flow over the eastern Sahel (not shown). Around 20 June, the SHL abruptly weakens and shifts



to a more southerly position, followed by a gradual intensification and northward retreat during the following week (Fig. 4a). It is still located east of its climatological position during this period (Fig. 4b). After that, a long strong phase begins to last until July 18 (Fig. 4a), during which the SHL gradually shifts westward (Fig. 4b) and also slightly northward. Around 18 July another abrupt weakening occurs and continues until the end of July 2016, only shortly interrupted by positive values

(Fig. 4a). The SHL is located to the west of its climatological position during this time (Fig. 4b) and then migrates to the south at the end of July (Fig. 4a). In the next section, the impact of these fluctuations and those of SSTs on synoptic-scale variability over SWA will be discussed.

## 4 Detailed synoptic analysis

### 4.1 General approach

In order to guide the discussion of the DACCIWA field measurements, the study period is divided into distinctive phases and the most significant weather systems are labelled for better reference in other papers. The division into phases is mainly based on the north-south precipitation difference (NSPD hereafter) between the coastal zone (0–7.5°N) and the Soudanian-Sahelian zone (7.5°N–15°N), both averaged across the longitude range 8°W–8°E (see Fig. 6 for orientation). Figure 5a shows daily values of the NSPD based on TRMM precipitation estimates for June-July 2016. Figure 5b shows the

corresponding zonally averaged rainfall values against latitude. Four distinct phases are recognisable from this analysis.
Phase 1 lasts from 01 to 21 June 2016 and is characterised by a rainfall maximum near the coast, however showing large fluctuations with periods around 5 days (Fig. 5a). Particularly the middle part of Phase 1 is very wet, while the earlier and later parts are characterised by more isolated rainfall peaks near 4°N (Fig. 5b). Rainfall during this period is unusually intense offshore of the Niger Delta area stretching across the Gulf of Guinea towards Cape Palmas (Fig. 6a, see also Fig. S1a

for anomalies). A second rainfall maximum is located over the tropical Atlantic to the west of West Africa in Fig. 6a, where SSTs are climatologically much warmer (not shown). Precipitation does already stretch far inland into the Sahel but amounts are relatively low with the exception of the Cameroon Line Highland region along the border of Nigeria and Cameroon. The pre-monsoonal conditions are also reflected in fields of zonally averaged total column water vapour (TCWV) with values above 45 mm mostly restricted to south of 12°N (Fig. 7). The ITD (identified by the 14°C-isoline of 2m dewpoint) fluctuates

around 16°N (Fig. 7).
Phase 1 corresponds closely to the period of anomalously strong and east–west fluctuating SHL discussed in sect. 3 (see Fig. 4), while correspondence to SST behaviour in the Gulf of Guinea (Fig. 3) is less clear. It is interesting to note that despite a strong SHL and an established ACT, rainfall remains strongest along the coast, indicating that monsoon onset has not yet occurred. This aspect will be discussed further in sect. 4.3. In Fig. 5, significant synoptic disturbances are labelled with

capital letters A–J. These were subjectively identified from 850-hPa streamline plots at 00 UTC each day and often show a noticeable correspondence to the precipitation and TCWV behaviour. A summary of their most important characteristics is given in Table 1 and individual tracks are shown in Figs. 11, 13, 14 and 16. The locations of the feature labels in Figs. 5 and



7 correspond to the times when they cross the Greenwich meridian (i.e. centre of the DACCIWA focus region) and their latitudinal position (Figs. 5b and 7 only).

The second Phase lasts from 22 June to 20 July 2016 and is characterised by a rainfall maximum inland with smaller and less regular fluctuations of the NSPD (Fig. 5a), and only occasional and weaker convective systems around 4°N (Fig. 5b). This indicates a fully developed WAM with a deeper penetration of rainfalls and TCWV into the continent, and a northward shifted ITD, while marine precipitation is restricted to the Gulf of Bonny and the waters along the West African west coast (Figs. 6b and 7). Large parts of the inland DACCIWA region were virtually dry during this period, much drier than in other years (Fig. S1b), despite relatively high TCWV values (Fig. 7). The transition from Phase 1 to Phase 2 is marked by strikingly dry conditions across most of the area of interest (Fig. 5b), much reduced TCWV (Fig. 7), strong fluctuations of the ITD (Fig. 7) and an abrupt breakdown of the SHL (Fig. 4a). During Phase 2 the SHL then gradually intensifies and shifts westward (Fig. 4). There is also a gradual increase of coastal upwelling during this period (Fig. 3), which is consistent with more stable, near-surface monsoonal winds. The behaviour of the ACT, which is relatively weak and shifted to the south during most of Phase 2, does not seem to be closely related to the precipitation shift.

During 21–26 July 2016 (Phase 3), the rainfall maximum shifts back to the coastal zone (Fig. 5a), accompanied by wet conditions spanning large parts of the latitude band from 1–22°N (Fig. 5b), where TCWV is enhanced and the ITD reaches its northernmost extension (Fig. 7). A horizontal distribution of rainfall during this period (Fig. 6c) shows unusually intense convection across the entire Gulf of Guinea, widespread rain across the entire Soudanian zone and more patchy local maxima stretching into the Sahel and even southern Sahara (see anomalies in Fig. S1c). Even larger amounts are found along the coast of Guinea and Sierra Leone. This wet phase is preceded and accompanied by a second breakdown of the SHL as well as a marked westward shift of its centre (Fig. 4). Coastal upwelling is increased during this period, while no major change to the ACT is seen (Fig. 3).

During the last 5 days of July 2016 (Phase 4), the WAM system returns to a more typical behaviour for this time of the year with a precipitation maximum in the Sahel, similar to Phase 2 (Fig. 5). As in Phase 2, the southern parts of the DACCIWA region are rather dry and coastal rainfalls are restricted to the Niger Delta region (Fig. 6d). Rainfall along the coast of Guinea, however, is even more abnormal than in Phase 3 (Fig. S1d). Overall, conditions are somewhat wetter than during Phase 2. This is accompanied by a partly recovery of the SHL (Fig. 4) and weakening of coastal upwelling (Fig. 3).

In the remainder of this section, the four Phases outlined above as well as the transition between Phases 1 and 2 (the monsoon onset) will be analysed in detail, focusing on the synoptic-scale features labelled in Fig. 5. To aid the characterisation of these features, the following additional diagrams will be considered (see sect. 2.2 for more details): (a) objective analyses of AEJ position and speed (Fig. 8), (b) Hovmöller plots of 850-hPa vorticity and meridional wind for the 4–18°N latitude band (Fig. 9) and (c) Hovmöller plots of equatorial wave disturbances in the 0–15°N band based on TRMM rainfall as well as tracks of long-lived MCSs (Fig. 10).





### 4.2 Phase 1: Pre-onset (01–21 June 2016))

As stated in sect. 4.1, the pre-onset period is characterised by a coastal rainfall maximum (Fig. 5), a strong, eastward shifted SHL (Fig. 4) and a weak ACT (Fig. 3). The AEJ is still located close to the coast during most of this phase (Fig. 8a, see also Fig. S2a). The first week (01–06 June) is relatively quiet with overall little rainfall across the region (Fig. 5b). The AEJ is

anomalously far south (Fig. 8a) with a below normal intensity (Fig. 8b). No significant coherent features are detected during this period, neither in 850-hPa vorticity and meridional winds (Fig. 9) nor in terms of filtered equatorial waves (Fig. 10). The enhanced vorticity feature starting on 05 June 2016 (Fig. 9) is related to a northern area of high horizontal wind shear (not shown) and thus is not associated with coherent meridional wind signals. The activity of long-lived MCSs is also relatively weak (black lines in Fig. 10, see also Fig. S3).

Between 07 and 15 June 2016 the AEJ begins shifting northward showing two distinct mean speed maxima of more than 14 m s$^{-1}$ (Fig. 8). On 07 June the jet maximum is located over southern Chad (not shown). The enhanced shear associated with this feature appears to have supported the formation of a large and long-lived MCS (Fig. 10) that brings substantial rainfall to southern areas (Fig. 5b) and thus creates a minimum in NSPD (Fig. 5a). In the following days, three relatively weak cyclonic disturbances cross the region (Fig. 11). As already mentioned, Table 1 provides a summary of main

characteristics of these disturbances and all subsequent ones. The first disturbance (labelled "A" in Fig. 11) propagates quickly westward from eastern Chad to northern Ivory Coast between 9 and 11 June 2016, in accordance with the relatively strong AEJ during this period (Fig. 8b). When it passes the DACCIWA region, the increase in southerly flow seen in Fig. 9 (solid black lines) is associated with an increase of rainfall inland, while coastal rainfall is also still active, leading to an NSPD near zero (Fig. 5a). The vorticity signature of Feature A is relatively weak (Fig. 9), but there are several long-lived

MCS embedded in this system and the latter stages are identified as a TD (Fig. 10).

The immediately following second disturbance (labelled "B" in Fig. 11) propagates a little slower and on a more southern track from the border of Nigeria and Cameroon parallel to the coast out to the Atlantic past 20°W. When the centre of the vortex passes the DACCIWA region on 12 and 13 June, a strong increase in rainfall over the ocean is observed, creating a sharp minimum in NSPD (Fig. 5a). The slower propagation of this feature is consistent with the larger distance to the strong

AEJ core near 9°N (Fig. 8). Feature B shows a more coherent signature in vorticity and meridional wind (Fig. 9), as well as TCWV (Fig. 7), and is identified as a TD with two very long-lived and intense MCSs embedded over the DACCIWA region (Fig. 10).

From 15–18 June 2016 the AEJ is weak and shifts northward (Fig. 8), while a third cyclonic feature becomes evident in the 850-hPa streamlines (labelled "C" in Fig. 11). It propagates relatively slowly from eastern Nigeria across the DACCIWA

region, reaching northwestern Ivory Coast by 00 UTC on 18 June. It is associated with a moderate increase in rainfall inland, while the coast is conspicuously dry, leading to a slightly positive NSPD (Fig. 5a). Interestingly, Feature "C" appears to be related to a longer-lived, somewhat patchy vorticity and meridional wind feature (white line in Fig. 9), which is also identified as a TD (Fig. 10). The vortex identified from the streamlines appears to move somewhat slower than this




disturbance and is only found during the period of strongest southerlies immediately over the DACCIWA region (not shown). The vorticity feature moves at a similar speed as the embedded MCS (Fig. 10). Interestingly, during the middle part of Phase 1, the rainfall in the 5–15°N latitude band appears to be modulated by two Kelvin waves propagating across the DACCIWA region (green lines in Fig. 10), which superpose with the TD signals. There is also some indication for an

equatorial Rossby wave activity in the western part of the domain, but this signal is harder to see in the unfiltered TRMM data (Fig. 10).

**4.3 Transition from Phase 1 to Phase 2: The onset (16–26 June 2016)**

The monsoon onset is often defined as a more permanent shift of the rainfall maximum into the continent (e.g. Fitzpatrick et al., 2015). According to the NSPD (Fig. 5a), this occurred on 21–22 June in 2016, the transition from Phase 1 to Phase 2. As

this date is of such a large importance for the WAM, a dedicated discussion of the five days before and after this date are presented here. Overall this 10-day period has relatively low rainfall in the DACCIWA region, the two noteworthy exceptions being the enhanced coastal rainfall around 19 and 20 June and a Sahelian maximum on 24–25 June (Fig. 5b). Consistently, the activity of equatorial waves and long-lived MCSs is strongly suppressed (Fig. 10).

The synoptic development during the onset is characterised by substantial extratropical influences disturbing the circulation

over northern Africa with high-amplitude waves and wave breaking in the subtropical upper troposphere. On 15 June 2016, the polar and subtropical jets merge over the Mediterranean Sea and a high-amplitude ridge is located upstream over the central North Atlantic (not shown). While the Atlantic ridge continues to propagate eastward, the downstream trough and ridge amplify strongly until 17 June. On this day, the trough stretches all the way to the Mauritanian coast and leads to strong southwesterly flow at 600hPa across the western Sahara (Fig. 12). It is conceivable that subsidence associated with the ridge

stretching from eastern Europe into northeastern Africa (Fig. 12) has contributed to the suppression of rainfall evident from Fig. 5b. The inflow of cool maritime air from the Atlantic Ocean leads to an abrupt ventilation of the SHL, causing a weakening of its intensity and rapid eastward shift of the centre (Fig. 4) with some resemblance to the situation Todd et al. (2013) refer to as the "maritime phase". The extreme nature of this cold surge is visible in 20°W–0°E averaged temperature anomalies at 850 hPa, showing a very distinct and unusual cooling during this period with anomalies below –6K north of

30°N and below –4°K down to almost 20°N (Fig. S4a). Over the next few days, the whole wave slowly drifts eastwards allowing the northerlies associated with the trough to penetrate into the eastern parts of the Sahara as well. The associated cooling, which is overall less dramatic in the east (Fig. S4b), finally leads to the conspicuous collapse of the SHL between 20 and 25 June 2016 shown in Fig. 4a. This development is also reflected in an abrupt northward jump of the AEJ core accompanied by a significant weakening around 21 June 2016 (Fig. 8).

Within the large area of reduced surface pressure to the southeast of the unusually far southward stretching trough (Fig. 12), three cyclonic vortices form at 850 hPa (Fig. 13). The first one (labelled "$D_1$") first appears over the Hoggar Mountains in southern Algeria on 15 June 2016 and then remains rather stationary over northern Niger between 16 and 18 June 2016. The second (labelled "$D_2$") forms farther to the east and slowly moves along the border between Chad and Niger between 17 and




19 June 2016 (see also Fig. 12). The last one (labelled "$D_3$") is only discernable in 850 hPa streamlines on 19 and 20 June 2016 along the border of northern Chad and Sudan. On these two days, the three centres form a zonally elongated area of cyclonic rotation with marked northerlies to the west and southerlies to the east. Between 19 and 21 June 2016, $D_1$ and $D_2$ rotate cyclonically around each other while beginning to propagate westward in a fashion similar to an AEW (Fig. 13). Both

cyclonic centres slow down and weaken between 22 and 25 June 2016 near the West African west coast. To the northwest of $D_1$ and $D_2$ northerly flow reaches values of 15–25 m s$^{-1}$ between 18 and 21 June 2016 (not shown), which leads to a marked southward push of TCWV (Fig. 7).

Figure 9 shows how this unusual development is reflected in 850-hPa vorticity and meridional wind (label "D"). On 18 June 2016, $D_1$ and $D_2$ are still located to the north of 18°N and thus signals in the Hovmöller plot are weak. On 19 June 2016, the

strong northerlies begin to penetrate into the DACCIWA region, helping to suppress rainfall (see Fig. 5b). This is followed by unusually large vorticity values on 20 June 2016, when $D_1$ moves south of 18°N (Fig. 13). On 20 and into 21 June 2016 a wide area of very strong southerlies spreads across the DACCIWA region. These bring moisture far into the continent, shifting the ITD northward (Fig. 7), and thus create the conditions for an inland rainfall maximum between 23 and 25 June 2016, indicating that the onset has in fact occurred. After the turbulent transition phase, the WAM system becomes relatively

quiet and the AEJ slowly gets re-established near its climatological latitudinal position until 26 June 2016 (Fig. 8) with the SHL also beginning to re-intensify (Fig. 4).

The analysis above strongly suggests that in 2016, the monsoon onset was triggered by very strong interactions with the midlatitudes that supported a suppression of rainfall over West Africa. Low-rainfall conditions around the onset have been documented for other years as well (Sultan and Janicot, 2003).

**4.4 Phase 2: Post-onset (22 June–20 July 2016)**

Phase 2 comprises a period of relatively undisturbed monsoon conditions. The entire DACCIWA aircraft campaign fell into this period (see Flamant et al., 2017, submitted). The NSPD is positive through most of this phase and is modulated by the significant weather systems E–I with centres between 12 and 16°N and thus farther north than the Phase 1 features A–C (see Fig. 5). This period was anomalously dry along the Guinea Coast (Fig. S1b). Features E–I also modulate the speed and

latitude of the AEJ (Fig. 8).

At the very beginning of this phase, between 23 and 26 June 2016, while the monsoon is still being established, a relatively weak (and therefore unlabelled) cyclonic feature crosses the southern part of SWA, creating some moderate rainfalls in the Sahel around 24 June 2016 (Fig. 5b). This is the first time during the DACCIWA campaign that the precipitation maximum has fully shifted inland. After this system, the SHL starts intensifying and shifts north (Fig. 4a), the AEJ accelerates (Fig. 8b)

and a deep southwesterly monsoon flow gets established (not shown).

Between 27 June and 08 July 2016, three AEWs develop (Fig. 14a) associated with moderate fluctuations in TCWV (Fig. 7). The first two (Features E and F) form on a relatively well organised AEJ (Fig. 8b) and show rather classical propagation characteristics with coherent signals in meridionally averaged vorticity and meridional wind at 850 hPa (Fig. 9). Both have





two cyclonic centres at 850 hPa straddling the jet to the north and south (Fig. 14a) and are also objectively identified as TDs (Fig. 10). Feature E, which forms near the Greenwich Meridian, appears to be associated with the slight rainfall enhancement on 26 and 27 June 2016, while Feature F forming near 12°E creates a peak in rain in the northern box on 29 June 2016 (Fig. 5b). Finally, Feature G forms during a period when the AEJ weakens and becomes more fragmented, which makes its

latitudinal position vary strongly (Fig. 8). This leads to less clear and slower propagation behaviour (Fig. 14a). The northern centre propagates from central Niger to eastern Mauritania between 03 and 06 July 2016 and then drifts northwestward towards the border with Western Sahara. This behaviour is accompanied by a rapid shift of the SHL to the west (Fig. 4b). The southern centre shows a less coherent propagation. This is consistent with the relatively patchy signals in wind and vorticity shown in Fig. 9, apart maybe from the final stages over the open Atlantic Ocean. Feature G is also not matched with

a TD as the previous features (Fig. 10). Nevertheless a marked increase in rainfall is observed when this feature crosses the DACCIWA region on 04 July 2016 (Fig. 5b).

After that, between 09 and 16 July 2016, a fundamentally different and quite unusual development occurs. While in the north, a cyclonic feature slowly tracks from eastern Mali to Cape Verde between 08 and 13 July 2016 and then out to the Atlantic (Feature $H_1$ in Fig. 14b), there is no clear corresponding southern vortex. Instead, an anticyclonic system ($H_2$)

slowly propagates from Gabon on 11 July across the tropical eastern Atlantic, reaching the coast of Sierra Leone on 14 July 2016, after which it begins to weaken over the ocean to the west. As this system moves a little faster than its cyclonic counterpart to the north, the two centres approach each other, creating an area of marked low- to midlevel southwesterly winds in-between them, particularly on 12–14 July 2016 in the west / to the west of the DACCIWA region (arrows in Fig. 14b). This behaviour is associated with a weakening and northward shift of the AEJ (Fig. 8). It is conceivable that these

westerly wind anomalies have also helped to intensify coastal upwelling as shown in Fig. 3. Given the zonal distance between the two centres, both positive and negative vorticity signals are apparent in the Hovmöller plot shown in Fig. 9, although the negative one is only strong past 10°W. Propagation of these two features is relatively slow with about 7 m s$^{-1}$. While the signal in the northerlies at 850 hPa is somewhat patchy, the signal in the southerlies, created by the positive superposition of the wind disturbances associated with the staggered northern and southern vortices, is coherent and strong,

particularly to the west of 10°W (Fig. 9), as also reflected in TCWV (Fig. 7). This has likely supported a deeper inland penetration and slight intensification of rainfall (Fig. 5b). Given the somewhat unusual behaviour of this system, it is no surprise that there is no matching between the TDs and long-lived MCSs objectively identified during this period (Fig. 10). To the best of our knowledge there is no description in the literature of a propagating cyclonic-anticyclonic vortex couplet unrelated to any of the classical equatorial waves. Thus, the dynamical origin of this feature is unclear.

An interesting effect on the coastal region is that the southern anticyclonic vortex (Feature $H_2$ in Fig. 14b) appears to have brought with it dry air from the area of subsidence in the equatorial zone or even SH. To illustrate this, Fig. 15 shows a time series of radiosoundings made four times daily from Abidjan for the period 07–16 July 2016. While most days show a well-developed monsoon layer with high relative humidity and winds from westerly directions, very dry air suddenly intrudes into the 850–700 hPa layer on midday of 11 July (drop from ~85 to under 20%) persisting until the morning of 14 July 2016.





Winds blow from southwesterly to westerly directions during this period. Indications of aged aerosol particles were found when the DACCIWA research aircraft penetrated this layer over several days, likely from fires in the SH (Flamant et al., 2017, submitted). Interestingly the usually easterly 600-925 hPa shear vector backs to northerly just before the dry event – signalling the changes in the 600 hPa circulation due to the vortex couplet.

**4.5 Phase 3: Wet westerly regime (21–26 July 2016)**

Phase 3 is characterized by wet conditions stretching from the tropical Atlantic far into the Sahel and even southern Sahara, particularly on 23–25 July 2016 (Figs. 5). Rainfalls are most abundant over the ocean, particularly off the coast of Nigeria and stretching west to Ivory Coast as well as off the coasts of Liberia and Guinea (Fig. 6) creating large positive anomalies (Fig. S1c). There is also a marked Sahelian band that dips south into Ghana. The relatively large rainfall over the ocean

coincides with an area of enhanced meridional SST gradients (Fig. 3) that influence surface wind convergence. Within the DACCIWA region (8°W–8°E) rainfalls to the south of 7.5°N dominate, leading to a sharp drop of NSPD to negative values during this period (Fig. 5a). Similar to the monsoon onset period during the transition from Phase 1 to Phase 2, Phase 3 is characterised by a breakdown of the SHL (Fig. 4a), but this time the responsible cold intrusion from the midlatitudes occurs over northeastern Africa (Fig. S4b), leading to a marked westward excursion of the SHL (Fig. 4b). Triggered by an upper-

wave in the subtropics, an intensification of the low-level northeasterlies is observed, particularly on 23 and 24 July 2016 over Egypt and Sudan. Such a situation has been referred to as a cold surge by Vizy and Cook (2009) and this type appears to be more frequent than the event in the west before the onset (see sect. 4.3).

An unusual and interesting synoptic development leads up to this event. On 17 July 2016, an anticyclonic centre appears over the northern Central African Republic and swiftly propagates westward, reaching the border between Nigeria and Benin

on 19 July 2016 (propagation speed ~12 m s$^{-1}$; Feature $I_2$ in Fig. 16). Strong southerlies ahead of this system lead to an increase in TCWV (Fig. 7). After this day, the vortex slows down substantially (average propagation speed ~4 m s$^{-1}$) and shifts to a more southern track just off the Guinea Coast and then out to the open Atlantic, reaching 25°W on 27 July 2016. During this period the vortex centre is not always clearly identifiable. Somewhat similar to Feature H (see Fig. 14b), the anticyclonic vortex is accompanied by a cyclonic centre to the north, which is first evident in streamlines at 850 hPa on 19

July 2016 close to the border between northeastern Nigeria and Chad (Feature $I_1$ in Fig. 16). During the slow propagation phase from 21–26 July 2016, when the two centres are almost aligned latitudinally with a distance of about 10 degrees, a strong westerly jet develops between them with a maximum near 10°N, which also propagates westward. The occurrence of this jet (see arrows in Fig. 16), which brings large amounts of moisture into SWA from the west, where much warmer SSTs prevail, exactly marks the beginning and end of Phase 3. Nicholson (2009) and Nicholson and Webster (2007) have shown

that summers with a strong westerly flow at 850 hPa are on average associated with particularly wet conditions. The situation discussed here therefore is one possible synoptic-scale manifestation of this climatological result. Finally, southerlies to the east of $I_1$ lead to a northward extension of the moist zone and unusual ITD position at 24°N (Fig. 7).





As with Feature H, the increasing westerly flow between the cyclonic and anticyclonic centres leads to a weakening and north- and eastward shift of the AEJ (Fig. 8, Fig. S2c) as well as to an increase in coastal upwelling (Fig. 3). The weakening of the AEJ may also be related to the weakened SHL discussed above. However, in contrast to Feature H, the latitudinal alignment of the vortices leads to a cancellation of signals in the Hovmöller diagram, making Feature I barely detectable in

Fig. 9. So unusual is this situation that there is also very little in terms of objectively identified wave features during this period (Fig. 10). However, some modulation of rainfall by an eastward propagating Kelvin wave is evident between 18 and 22 July 2016. Mounier et al. (2008) also discuss enhanced westerly inflow, moist conditions and a Kelvin wave influence in connection with what they term the quasi-biweekly zonal dipole but the match with their concept is hard to establish for a single case.

**4.6 Phase 4: Recovery (27–31 July 2016)**

On 26 July 2006 the widespread rainfall characterising Phase 3 ceases and the precipitation maximum shifts back to the Sahel for the rest of the DACCIWA campaign period, as indicated by a positive NSPD (Figs. 5 and 6d). There is also evidence for a return of the SHL and the ITD to more climatological positions and intensity (Figs. 4 and 7). So overall this phase marks the return to more undisturbed monsoonal conditions similar to Phase 2. A last significant cyclonic feature is

detected during this period (Fig. 16, labelled J). This feature is first detected over South Sudan on 23 and 24 July 2016. Until 27 July it swiftly crosses the DACCIWA region, reaching the Guinea Highlands. After that, it slows down over the Atlantic on 28–30 July 2016. Figure 8 shows that the period of fast propagation is concomitant with an enhanced and southward shifted AEJ. Feature J can be identified well in the north–south averaged 850-hPa vorticity in the eastern and western parts of the study region but is somewhat diffuse around 10°W (Fig. 9). Nevertheless the fast propagation phase is evident from

the vorticity as well. Meridional wind signals associated with Feature J, however, are rather weak and only the western parts are concomitant with an objectively identified TD, which in turn appears to be related to a long-lived MCS (Fig. 10). Feature J also creates some mild fluctuations in TCWV (Fig. 7).

**5 Impact on low clouds, dust, biomass burning aerosol and city pollution dispersion**

After the detailed discussion of the large-scale settings in sect. 3 and the synoptic evolution in sect. 4, this section aims to

discuss the impact of these variability patterns on low clouds and atmospheric composition, two particular scientific interests of the DACCIWA project (Knippertz et al., 2015a). Evaluating the behaviour of low clouds is difficult over summertime SWA due to a relatively sparse observational network at the surface and regular obscuring by mid- and high-level clouds (van der Linden et al., 2015). Figure 17 shows the fraction of low clouds (defined here as cloud top pressure of 800 hPa and lower) daily at 00 UTC in the 5–10°N, 8°W–8°E box (see Fig. 1) as analysed from the CLAAS-2 dataset (see sect. 2). This

fraction is relative to the number of pixels not obscured by higher clouds, which can be as high as 85% (red curve in Fig. 17), indicating a large uncertainty in the low-cloud estimate. Despite this, a clear difference in low-cloud cover between





the four Phases can be seen. Typical fractions during Phase 1 range around 60%. Given the large uncertainty, day-to-day variations should be regarded with caution and thus it is no surprise that the impact of individual synoptic features is generally hard to discern. Merely the wet Feature A and dry Feature D stand out with a high fraction of obscuring high clouds and a low fraction of low clouds for Feature A and the opposite for Feature D. During Phase 2 fractions of low clouds

typically range around 85%, while the very variable fraction of obscuring high clouds is on average a little lower than in Phase 1 (Fig. 17). This suggests that the onset is a prerequisite for the occurrence of the extensive stratus decks in SWA. Again, a clear influence of the synoptic features E–I is hard to discern. During Phase 3, the wet 24 July 2016 stands out as a day with a very high fraction of obscuring clouds and a very low (but also very uncertain) fraction of low clouds. Finally, Phase 4 returns to the more typical abundant low and less frequent high clouds, also observed during Phase 2. During

Phase 1, there is frequent evidence for a land-sea breeze convergence creating clouds in a line parallel to the coast around midday, at least in areas away from deep convection (not shown). Such behaviour is much more difficult to detect during the cloudier Phases 2–4.

Figure 18 shows vertically integrated fields from the ECMWF CAMS-IFS analysis, again averaged zonally from 8°W to 8°E. The loading of mineral dust is given as DAOD at 550 nm (Fig. 18a), while the impact of biomass burning is indicated

here through the vertically integrated number of CO molecules per surface area (Fig. 18b). With respect to dust, differences between the four phases are again evident. Before the onset, the dust plume stretches farther south and sometimes even reaches the coast (around 6°N). In particular, Features C and D create significant southward excursions of the dust plume, in contrast to the relatively weak southern disturbance B and the fast-propagating disturbance A. With the onset in Phase 2, the dusty zone retreats to the north of 8°N with visible modulation by all four cyclonic features E, F, G and H. The area of

southwesterlies between the cyclonic vortex $H_1$ and the anticyclonic vortex $H_2$ (see Fig. 14b) pushes the dust northwards. A few days later, a similar but even stronger northward push to beyond 15°N is evident in the aftermath of Feature I, creating a relatively dust-free Phase 3. Finally a return to conditions similar to Phase 2 occurs with the arrival of Feature J and throughout Phase 4.

For CO (Fig. 18b), the dispersal of the SH biomass burning plume northwards generally covers a wider latitudinal range and

the modulation by the four phases and the ten synoptic features is not quite as clear as for the dust. During Phase 1 considerable CO reach as far north as the Sahel, particularly in the area of southerlies following the passage of the centre of Feature A (see also Fig. 9). The more southern Feature B also instigates a northward transport, but this does not reach 10°N. Finally, Feature C is associated with a latitudinally extended but somewhat weaker CO plume. Comparing the two panels of Fig. 18 shows that during most of Phase 1 dust and biomass burning signatures co-existed in the vertical column over SWA,

particularly in the latitudinal range 8–10°N. The arrival of Feature D associated with the monsoon onset pushes higher CO values back into the SH, with the exception of a small plume around 20 June 2016. After the onset, higher CO slowly return to the DACCIWA region, reaching a peak northward extent to 12°N around 02 July 2016. For the rest of Phase 2 and until the arrival of Feature I, CO retreats to the coastal and oceanic areas. Remarkably the time after the passing of Feature H, which is associated with an anticyclonic vortex from the SH, does not show outstanding vertically integrated CO





concentrations. The DACCIWA aircraft frequently measured aged aerosol particles during this period. Further study is needed to check where this aerosol comes from and how deep the layer was vertically using the detailed field measurements. A marked increase in CO is found to occur with the arrival of Feature I, leading up to the highest values during the entire 2-months period according to CAMS-IFS. It appears that this plume is transported into the region around the anticyclonic

feature $I_2$ (Fig. 16). Finally, towards Phase 4, Feature J is associated with a return to values similar to the middle of Phase 2. It is interesting to note that during Phases 2–4, there is generally very little overlap between the vertically integrated dust and CO fields (cf. Figs. 18a with b) in contrast to Phase 1. During most of this time, there appears to be a narrow, meandering, relatively clean strip of air with dust to the north and CO to the south.

Finally, the dispersion of urban pollution plumes from the five cities of greatest interest to DACCIWA (Abidjan, Kumasi,

Accra, Lomé and Cotonou) is discussed based on FLEXPART and HYSPLIT results (see sect. 2.2). Figure 19 illustrates the spatial distribution of particles emitted and dispersed during Phases 1 and 2 simulated by the FLEXPART model. During Phase 1, when the monsoon flow is less established, pollution dispersion is more local and mostly directed into north- to eastward directions. During Phase 2, plumes are more clearly concentrated around the northeastward direction and stretching over long distances. Given the geographical distribution of the cities, Lomé and Cotonou are likely impacted by emissions

from Accra during Phase 2, and Kumasi from emissions from Abidjan. Evidently quite remote areas can be affected by pollution from coastal cities in relatively short time. The supersite in Savé for example, is close to the main Accra and Lomé pollution plumes. Whether these upwind plumes actually degrade air quality in receptor cities requires further exploration using ground-based and aircraft measurements from the DACCIWA field campaign. During the five days period of Phase 3, due to the increased westerlies between the cyclonic and anticyclonic centres $I_1$ and $I_2$ (Fig. 16), the dominant direction

shifted from northeast to east-northeast, while Phase 4 is more similar to Phase 2 (not shown).

Figure 20 summarizes the results for Phase 2 in the form of pollution roses for both models, giving some estimate for typical uncertainties in the dispersion estimates. As expected, the dominant transport direction is northeastward for both models and all cities, but some details clearly differ. For example a considerable fraction is transported east-northeastward for Accra in the HYSPLIT simulations and north-northeastward for Abidjan in FLEXPART. City pollution plumes generally reach a

distance of around 300 km from source point over the course of the 24-hour simulations, with the exception of shorter plumes from Abidjan (~200km) in the FLEXPART simulations.

**6 Conclusions**

Atmospheric variability over West Africa in summer is controlled by a wide range of different factors reaching from global SST patterns to local convection. Here we analysed these factors and their interplay exemplarily for the period of the main

DACCIWA field campaign, i.e. June-July 2016, on the basis of ECMWF model products, and satellite and radiosonde data. The DACCIWA campaign fell into a period of Pacific La Niña and Atlantic El Niño conditions, which statistically have opposing effects on Sahel rainfall. Mediterranean and Indian Ocean SSTs pointed towards a wetter than normal Sahel, but





most likely the relatively warm SSTs in the tropical Atlantic dominated and led to near-normal rainfalls across the whole of West Africa during 2016.

In order to better characterise the observed changes on a day-to-day basis, objective analysis of tropical wave features and tracking of long-lived MCSs were used together with a subjective tracking of vortices in 850-hPa streamlines. A summary of the latter is provided in Table 1. This analysis shed new light into the richness of synoptic-scale features affecting the region and their impacts on wind, precipitation, cloudiness, and the distribution of dust and biomass burning plumes. It serves as valuable background information for the more detailed examinations of the comprehensive dataset collected during the DACCIWA campaign from ground stations, aircraft, radiosonde and satellite and as inspiration for a deeper analysis of the under-researched propagating synoptic systems of SWA and their dynamics. The 2-month DACCIWA period can be divided into four distinct Phases:

- Phase 1 (01–21 June 2016): This period is characterised by pre-onset conditions with the rainfall maximum close to the Guinea Coast. The ACT becomes established in the course of this period, but coastal upwelling is still weak. The SHL is relatively intense with large east–west fluctuations, also accompanied by large variations in AEJ speed. Three relatively weak westward propagating vortices (A, B and C) affect the region, which are associated with TDs and long-lived MCSs, creating marked variations in precipitation. Typical coverage with low clouds over SWA during this Phase is about 60%. Mineral dust from the Sahara/Sahel penetrates far south and occasionally reaches the coast in significant concentrations, while at the same time high values of CO, used here as an indicator of biomass burning activity, push inland from the south. The weaker monsoon flow leads to less pollution dispersion from coastal cities. Towards the end of the Phase, strong influences from the extratropics occur in the form of a deep trough and cold surge over the western Sahara around 17 June 2016, leading to a collapse of the SHL and AEJ. A cyclonic disturbance (labelled D) with two centres develops, creating an area of strong, dry, dusty northerlies to the west, followed by a deep penetration of southerlies and re-moistening of the area, which finally creates the monsoon onset. This period is also characterised by a conspicuous absence of tropical wave features.

- Phase 2 (22 June–20 July 2016): The post-onset Phase 2 is characterised by a gradual re-intensification and westward shift of the SHL with an AEJ close to climatological position and speed for most of the period. The rainfall maximum has permanently shifted inland with an anomalously dry Guinea coastal region. Rainfalls and AEJ speed are being modulated by five significant synoptic-scale features E–I. While the first three show characteristics of classical AEWs, the latter two consist of a northern cyclonic and southern anticyclonic centre. For Feature H, the anticyclonic centre is shifted eastward and slowly moves north from the SH, bringing with it a shallow layer of dry air filled with aged aerosol. For Feature I, the anticyclonic centre has an origin over central Africa and is more aligned meridionally with its cyclonic counterpart. Together they create a more westerly near-surface flow accompanied by peaks in coastal upwelling. During Phase 2, low-cloud cover over SWA generally increases to about 85%. Mineral dust retreats to the Sahel and Soudanian zones with clear modulations by the major synoptic-scale features, while CO fields show a marked peak around 02 July 2016 and then weaker values afterwards. Pollution dispersion from coastal cities is stronger and mostly towards NNE or NE.





- Phase 3 (21–26 July 2016): The transition between Phases 2 and 3 is accompanied by a moderate cold-air intrusion into northeastern Africa. This is associated with a second breakdown of the SHL, and northward shift and weakening of the AEJ. The short Phase 3 itself is then characterised by overall wet conditions stretching from the tropical Atlantic into the southern Sahara with a maximum in the coastal zone. The main reason for this appears to be the moisture transport associated with the strong westerly flow between the cyclonic and anticyclonic centres of Feature I already starting in Phase 2. One further cyclonic feature (J) occurs during this period and modulates wind and rainfall. Notably, the wet Phase 3 is almost dust-free, but high column-loadings of CO penetrate deep into SWA. Pollution dispersion from coastal cities is strongest and mostly towards the northeast or even east-northeast.

- Phase 4 (27–31 July 2016): The final five days of the DACCIWA period are characterised by a return to more undisturbed monsoonal conditions with a more climatological SHL, AEJ and rainfall distribution. Dust, CO and city pollution plumes also return to conditions similar to Phase 2.

This analysis demonstrates the significant range of features affecting SWA around the period of the monsoon onset with marked impacts on cloudiness, rainfall, wind and pollution transport. Four types of behaviours can be distinguished: Particularly before the onset, but also during the re-establishment of the monsoon at the end of Phase 3, single cyclonic vortices occur at different latitudes with different propagation speeds (Features A, B, C and J). These are typically related to TDs and long-lived MCSs, but the exact dynamical reason for their existence is not entire clear (see also Fig. 4 in Schrage et al., 2006 for other examples). The second type is classical AEWs with a northern and southern cyclonic vortex (Features E, F and G). These have been described extensively in the literature and their dynamics are well understood (e.g. Hall et al., 2006). They usually have a discernable signal in vorticity, wind and precipitation fields and are also objectively identified as TDs. The third type, which appears to be rarer and whose climatological and dynamical characteristics are barely covered in the literature are jointly propagating cyclonic and anticyclonic vortices (Features H and I), which create an anomalous westerly flow in-between them, associated with enhanced coastal upwelling and a more eastward transport of city pollution plumes. The resulting conditions appear to depend on the exact origin of the involved air masses. If the strong westerly flow taps into moist air off the west coast, where high SSTs are common, this can lead to anomalously moist conditions across the region, as has been described climatologically by Nicholson (2009) and Nicholson and Webster (2007). Hand-analysed surface weather charts from Phase II of GATE show couplets of cyclonic-anticyclonic vortices, but no details are discussed (Sadler and Oda, 1979). Finally, extratropical influences can markedly impact on weather conditions over SWA. This is usually associated with midlatitude troughs penetrating into the Sahara, cold surges and disruptions to the SHL and AEJ. During the DACCIWA period, Feature D is of particular interest, as it appears to have caused a substantial dry anomaly before the actual monsoon onset. Such behaviour has been described in the literature (e.g. Sultan and Janicot, 2003), but the role of the extratropics in a given year is yet to be explored, ideally also with model sensitivity experiments. Feature D is also striking, since it transforms from a rather stationary low-pressure zone with several centres downstream of an extratropical trough into a westward propagating, much more tropical-like vortex couplet that has some resemblance with an AEW. A detailed analysis of the dynamics of this transition is beyond the scope of this study and left for future work. It is



noteworthy that periods of extratropical influences appear to create the most persistent meridional flow anomalies, leading to extreme excursions of mineral-dust and biomass-burning plumes over SWA.

In the future, it would be desirable to study the four characteristic types of variability patterns described above in a climatological (comparing 2016 to other years) and dynamical sense. The latter could be achieved both through theoretical

5   work on linear tropical modes taking into account the specific conditions over SWA in summer or possibly in a full non-linear sense through idealised model experiments or realistic case studies.

**Acknowledgments**

The DACCIWA project has received funding from the European Union Seventh Framework Programme (FP7/2007-2013) under grant agreement no. 603502. AS has been supported from subproject "C2 – Prediction of wet and dry periods of the

10   West African monsoon" of the Transregional Collaborative Research Center SFB/TRR 165 "Waves to Weather" funded by the German Research Foundation (DFG), MG by the LABEX project funded by Agence Nationale de la Recherche (French National Research Agency, grant ANR-10-LABX-18-01), and TB by BMBF Grant no. 01LP1520D (MIKLIP Promisa). The authors gratefully acknowledge the NOAA Air Resources Laboratory (ARL) for the provision of the HYSPLIT transport and dispersion model and READY website (http://www.ready.noaa.gov) used in this publication. The AERIS/SEDOO data

15   infrastructure provided access to the GIRAFE/FLEXPART simulations, CAMS-IFS forecasts and data used in this study (http://www.aeris-data.fr and http://dacciwa.sedoo.fr). The GIRAFE/FLEXPART simulations were provided by Alain Fontaine (SEDOO). The authors would also like to thank Gregor Pante for help with producing Figs. 12 and 15 and Robert Redl for his development of the AEJ detection tool.



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



**Tables**

| LABEL | TIME | LON | DESCRIPTION |
|---|---|---|---|
| A | 09–12 June | 22°E – 25°W | fast propagating (16.8 m s$^{-1}$) Soudanian (~11°N) cyclonic vortex / tropical disturbance, weak vorticity signal but clear southerly wind signal, long-lived MCSs embedded |
| B | 12–15 June | 09°E – 21°W | moderately fast (11.2 m s$^{-1}$) coastal (~5°N) cyclonic vortex / tropical disturbance, moderate vorticity and meridional wind signals, intense, long-lived MCSs embedded |
| C | 15–18 June | 11°E – 08°W | moderately fast (11.0 m s$^{-1}$) Soudanian (~9°N) cyclonic vortex / tropical disturbance, patchy vorticity and meridional wind signals, long-lived MCS embedded |
| D | 15–25 June | 17°E – 23°W | low-pressure trough turning into westward propagating disturbance with two cyclonic centres, large meridional wind anomalies, relative dryness, triggering monsoon onset |
| E | 27–30 June | 02°E – 27°W | AEW with two cyclonic centres and typical propagation speed of 9.6 m s$^{-1}$, with coherent signals in meridional wind and vorticity, leading to increased Sahelian rainfall |
| F | 29 June – 03 July | 14°E – 21°W | AEW with two cyclonic centres and fast propagation speed of 10.6 m s$^{-1}$, with coherent signals in meridional wind and vorticity, leading to increased Sahelian rainfall |
| G | 03–08 July | 12°E – 18°W | unorganised AEW with ill-defined southern centre, northward moving northern centre and varying propagation speed, but discernable rainfall signal |
| H | 09–16 July | 12°E – 22°W | slowly moving (7.1 m s$^{-1}$) northern (~15°N) cyclonic and southern anticyclonic (~4°N) vortex originating from SH, westerly wind anomaly between centres |
| I | 17–27 July | 23°E – 25°W | relatively slow moving northern (~13°N) cyclonic and southern (~5°N) anticyclonic vortex with westerly wind anomaly in-between, creating conditions for wet period |
| J | 23–30 July | 19°E– 25°W | mostly slow moving Soudanian (~9°N) cyclonic vortex, coherent vorticity but less clear wind signal, occurring in an environment of MCSs, high moisture and widespread rain |

**Table 1: Characterisation of the labelled (Column 1) synoptic-scale features. Columns 2 and 3 give the time period and longitude range for which a coherent vortex was tracked in 850-hPa streamlines (see Figs. 11, 13, 14 and 16). Column 4 gives a general**
5   **description including aspects such as propagation speed, latitude and reflection in wind and vorticity anomalies.**





**FIGURES**

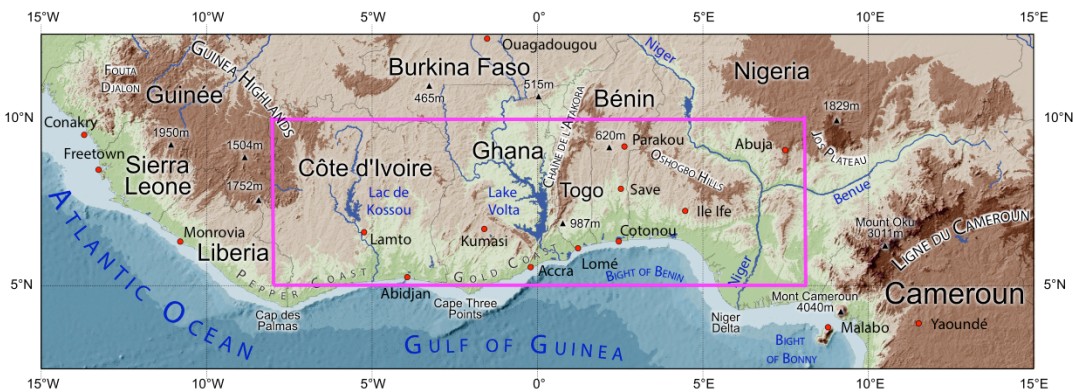

**Figure 1: Geographical overview of the study region with sites and names. The purple rectangle marks the main DACCIWA focus region (5–10°N, 8°W–8°E).**

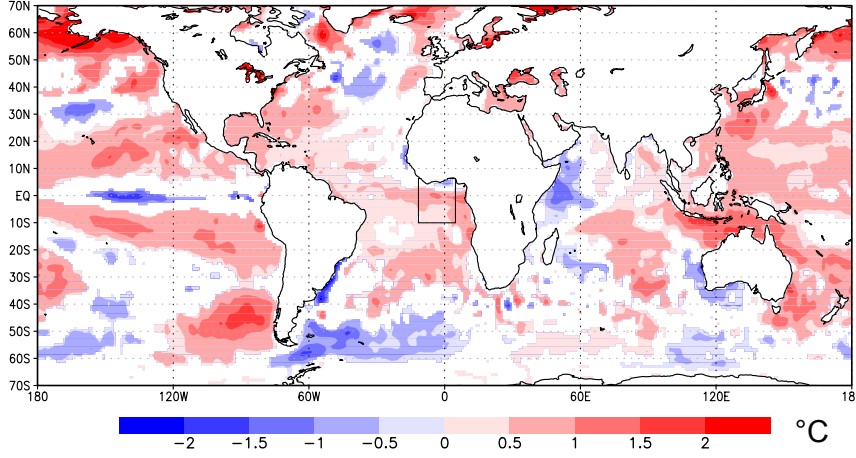

**Figure 2: Global SST anomalies for June-July 2016 [in °C]. Basis is the Optimal Interpolated Reynolds SST V2 dataset and**
10 **anomalies are relative to 1981–2016. Only anomalies above the 95% confidence level based on a two-sided Student's t-test are plotted. The black box marks the area used for Fig. 3.**





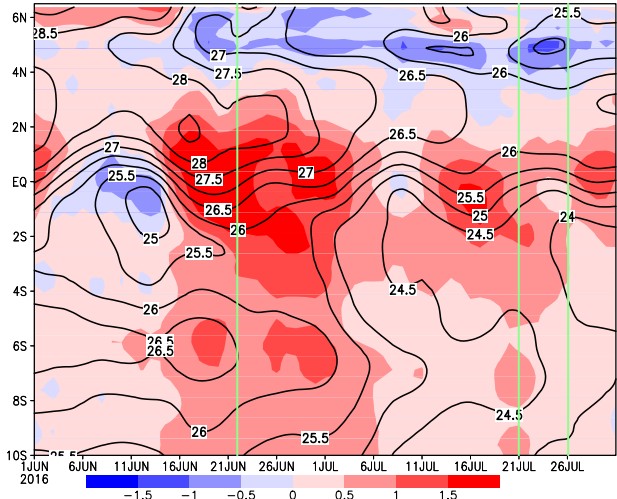

**Figure 3: Daily SST behaviour over the eastern tropical Atlantic during June-July 2016. SSTs [in °C] are averaged between 10°W and 4°E (see box in Fig. 2) and shown as absolute values (lines) and anomalies (shading). Basis is the Optimal Interpolated Reynolds SST dataset and anomalies are relative to 1981–2016 as in Fig. 2. Only values greater than SST Reynolds daily error estimation are plotted. The four Phases of the DACCIWA campaign are marked with thin green lines.**





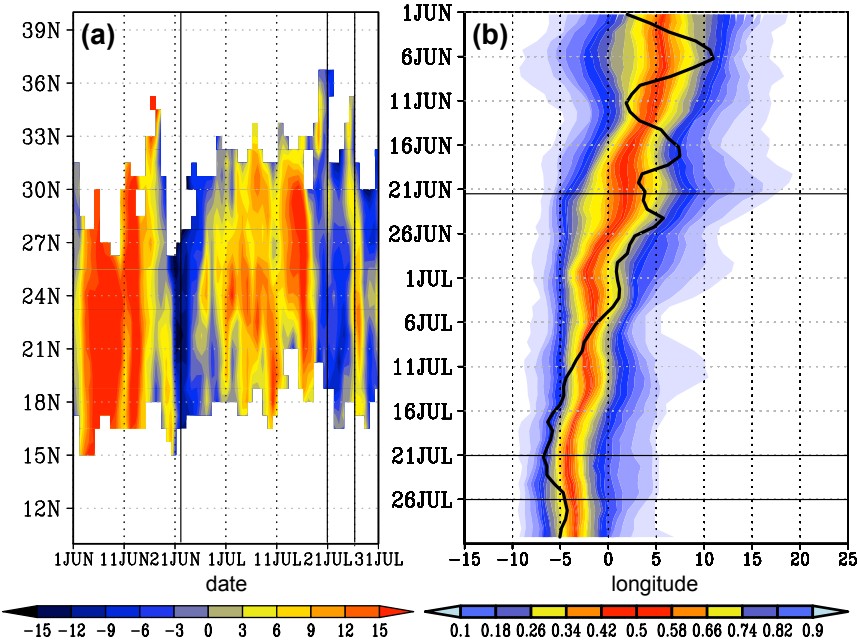

**Figure 4: SHL evolution during June-July 2016. (a) Time-latitude anomalies of the SHL intensity (in gpm) defined as the thickness between 925 and 700 hPa (relative to 1979–2016). (b) Longitudinal location of the SHL barycentre (black line) with the 1979–2016 percentiles in colour shading. See sect. 2.2 for more details. The four Phases of the DACCIWA campaign are marked with thin black lines.**





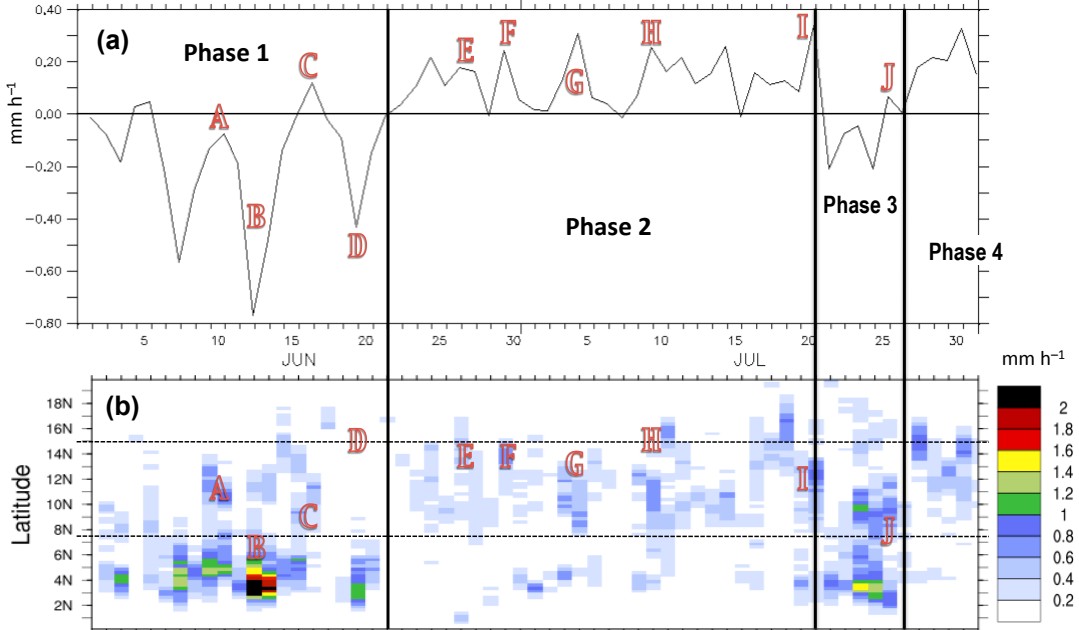

**Figure 5: Rainfall evolution during June-July 2016. (a) North–south precipitation difference based on the 7.5–15°N and 0–7.5°N bands (see boxes in Fig. 6). (b) Latitudinal distribution of rainfall. Both panels are based on daily TRMM precipitation values averaged over 8°W–8°E (longitudes bordering DACCIWA focus region, see Fig. 1). The four Phases of the DACCIWA campaign**
5 **and significant synoptic-scale features A–J are marked at the approximate time (and also latitude in (b)) of crossing the DACCIWA focus region.**



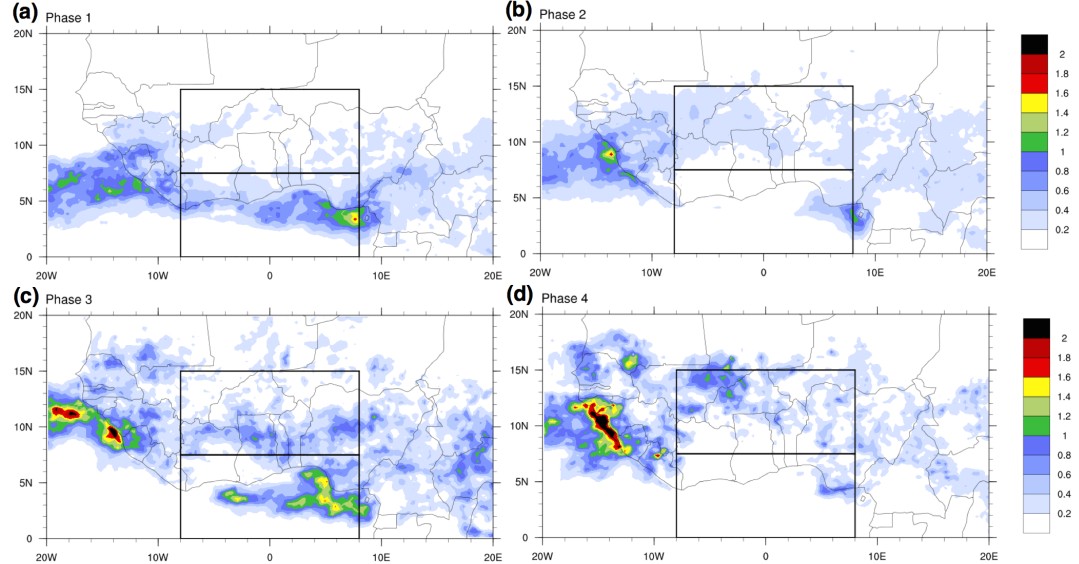

**Figure 6: Horizontal distribution of mean precipitation during the four Phases of the DACCIWA campaign. Plots are based on TRMM precipitation and given in mm h⁻¹. (a) Phase 1 (01–21 June 2016), (b) Phase 2 (22 June–20 July 2016), (c) Phase 3 (21–26 July 2016) and (d) Phase 4 (27–31 July 2016). The black boxes mark the areas used to compute the north–south precipitation difference shown in Fig. 5a. Corresponding anomalies are shown in Fig. S1.**

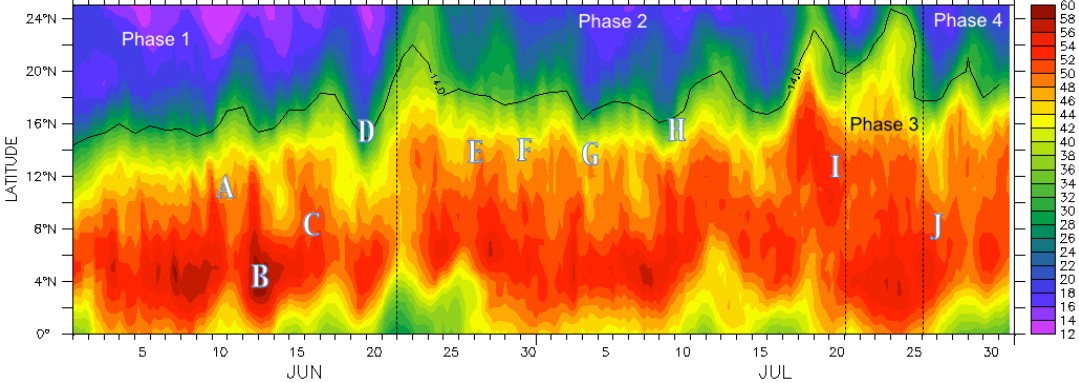

**Figure 7: Evolution of TCWV (shading in mm) and the ITD (black line, identified from the 14°C isoline of 2-m dewpoint) during June-July 2016 based on ECMWF operational analysis. The four Phases of the DACCIWA campaign and significant synoptic-scale features A–J are marked at the approximate time and latitude of crossing the DACCIWA focus region as in Fig. 5b.**





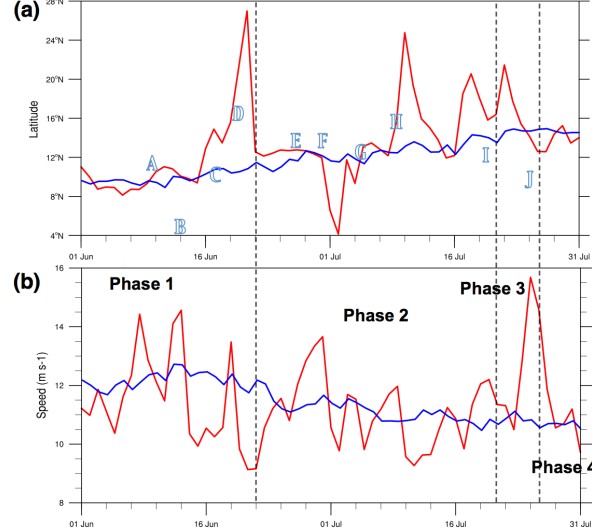

**Figure 8: AEJ evolution during June-July 2016. Time series of (a) latitudinal position and (b) mean speed in m s$^{-1}$ of the AEJ objectively identified from ERA-I re-analysis data. The red lines indicate the 2016 evolution, the blue lines the 1987–2016 climatological mean (see sect. 2.2 for more details). The four Phases of the DACCIWA campaign are indicated by vertical lines. The significant synoptic-scale features A–J are marked at the approximate time and latitude of crossing the DACCIWA focus region.**





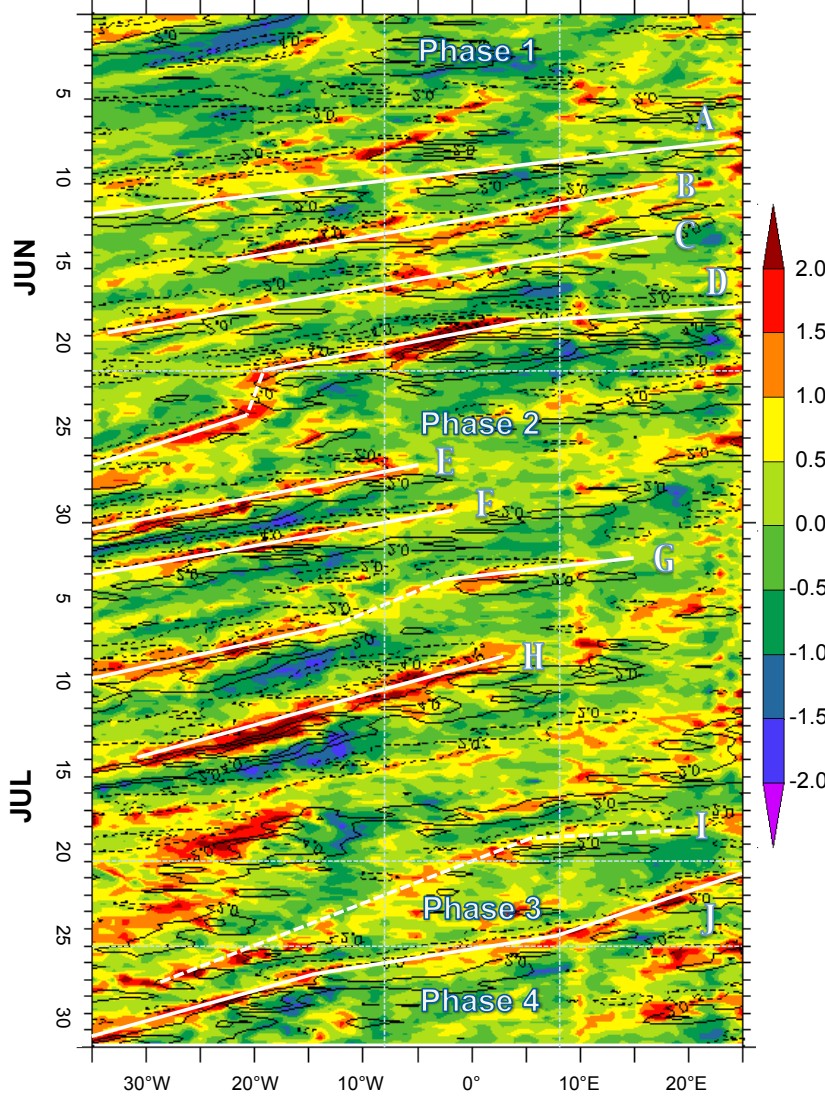

**Figure 9: Coherent wind and vorticity features affecting the DACCIWA region.** Hovmöller diagram showing 4–18°N meridionally averaged vorticity (colours, in $10^{-5}$ s$^{-1}$) and meridional wind (black lines, in m s$^{-1}$) based on operational ECMWF analyses in 1° resolution (in order to smooth noisy vorticity fields). The four Phases of the DACCIWA campaign and significant synoptic-scale features A–J are marked as well as the longitudinal bounds of the DACCIWA focus region 8°W–8°E (see Fig. 1).



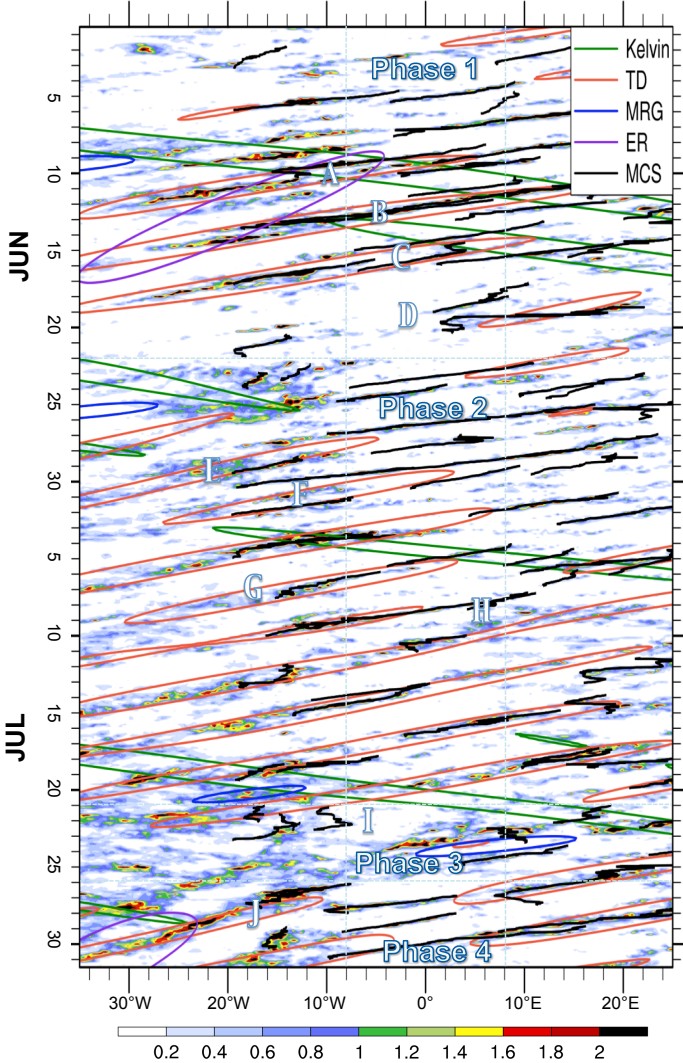

**Figure 10: Tropical wave phenomena and long-lived MCss during June-July 2016.** Hovmöller diagram of 5–15°N meridionally averaged precipitation from TRMM (in mm h$^{-1}$, colour shading according to legend) with objectively identified waves marked with coloured lines according to the legend in the top right corner and long-lived MCSs with at least 24 hours of lifetime marked with thick black lines (for details of detection of both features, see sect. 2.2). Contour lines for the wave features correspond to a modulation of precipitation of more than 0.12 mm h$^{-1}$. Note that while the tropical waves are identified for the entire longitudinal range of 35°W–25°E, the MCS identification is limited to the land-dominated area 20°W–25°E. The four Phases of the DACCIWA campaign and significant synoptic-scale features A–J are marked as well as the longitudinal bounds of the DACCIWA focus region 8°W–8°E (see Fig. 1).





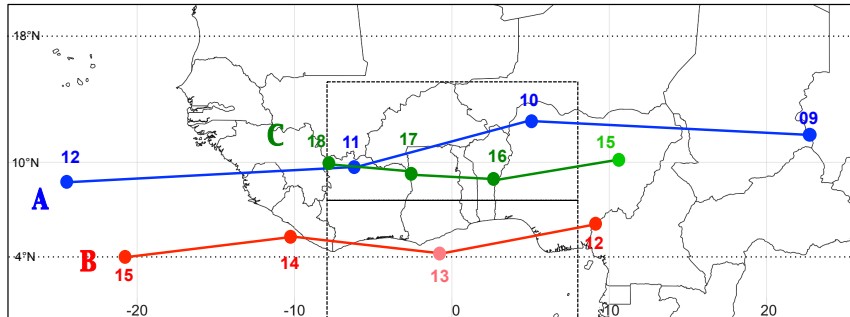

**Figure 11: Significant synoptic-scale features during 09–18 June 2016 (Phase 1). To create this graph, vortices were subjectively identified in 850-hPa streamlines based on operational ECMWF analyses. All vortex positions refer to 00 UTC with the date given as numbers. Round symbols mark cyclonic systems (labelled A , B and C). Paler colours are used for days, when the vortices were not clearly identifiable. The boxes mark the areas used to compute the NSPD shown in Fig. 5a. The stippled lines shows the latitude range used to produce Fig. 9.**

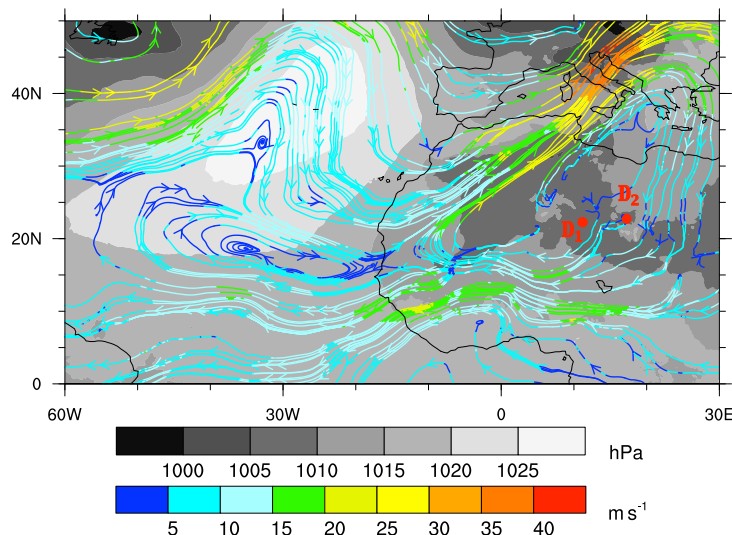

**Figure 12: Extratropical influences during the monsoon onset. Shown are streamlines coloured by wind speed (scale at bottom), at 600 hPa and mean sea-level pressure (grey shading) at 00 UTC 17 July based on ECMWF operational analyses. The two disturbances from Fig. 13 are marked in red.**





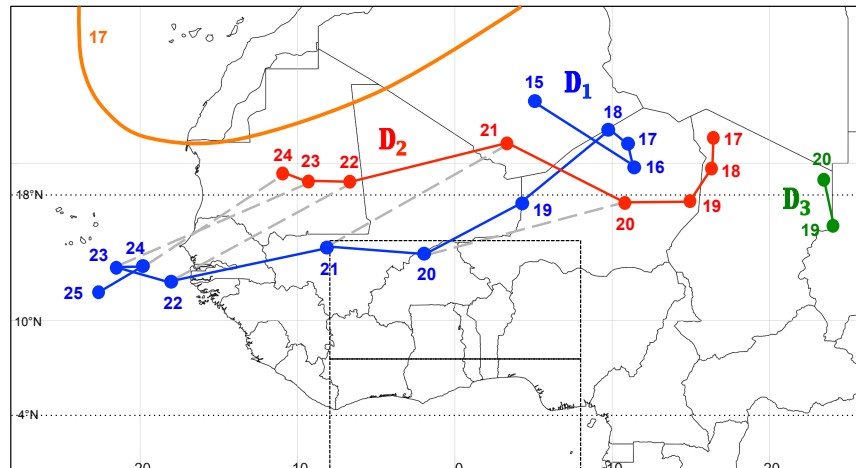

**Figure 13: Significant synoptic-scale features during 15–25 July 2016 (transition from Phase 1 to Phase 2). To create this graph, vortices were subjectively identified in 850-hPa streamlines based on operational ECMWF analyses. All vortex positions refer to 00 UTC with the date given as numbers. Round symbols mark cyclonic vortices (labelled $D_1$, $D_2$ and $D_3$) and the thick orange line the southernmost extension of a significant trough at 600 hPa (see Fig. 12). Vortices with a joint propagation are linked with dashed grey lines. The boxes mark the areas used to compute the NSPD shown in Fig. 5a. The stippled lines shows the latitude range used to produce Fig. 9.**




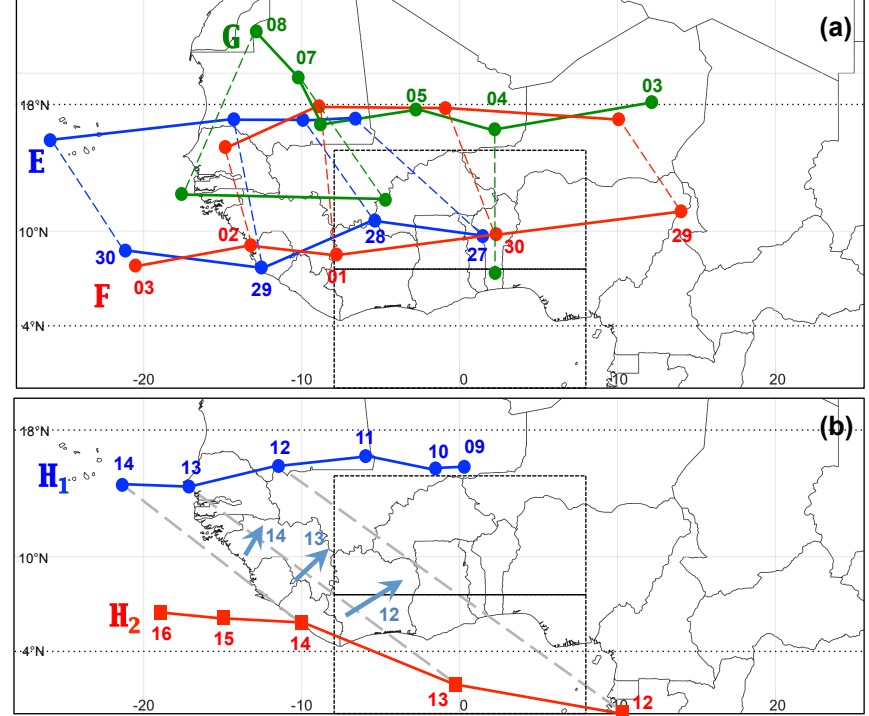

**Figure 14: Significant synoptic-scale features during 27 June–16 July 2016 (beginning (a) and middle (b) of Phase 2). To create these graphs, vortices were subjectively identified in 850-hPa streamlines based on operational ECMWF analyses. All vortex positions refer to 00 UTC with the date given as numbers. Round symbols mark cyclonic systems (labelled E, F, G and $H_1$),**
5 **squares anticyclonic systems (labelled $H_2$). Vortices with a joint propagation are linked with dashed grey lines. The cores of significant 850-hPa jets are indicated with light blue arrows, again with the date at 00 UTC given as numbers. The boxes mark the areas used to compute the NSPD shown in Fig. 5a. The stippled lines shows the latitude range used to produce Fig. 9.**





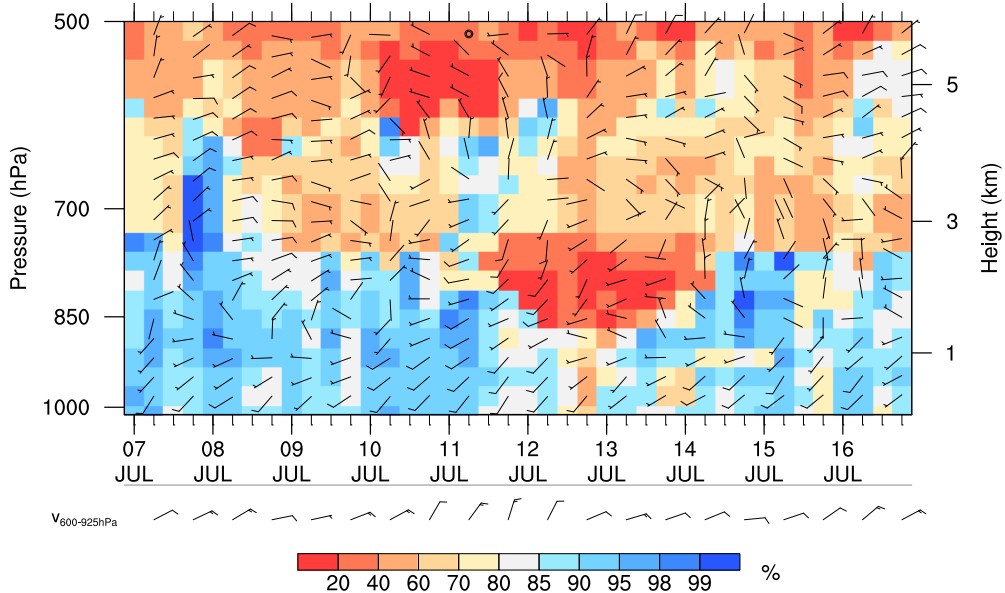

**Figure 15: Vertical structure of the atmosphere during part of Phase 2. Shown are relative humidity (shading according to scale, four times daily), wind (barbs) and 600–925-hPa vertical wind shear (below main plot) (both two times daily) from radiosondes launched at Abidjan (for location see Fig. 1) during 07–16 July 2016.**

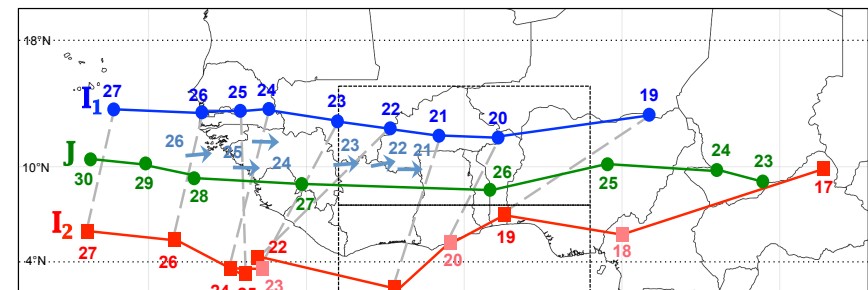

**Figure 16: Significant synoptic-scale features during 17–30 July 2016 (end of Phase 2 to Phase 4). To create this graph, vortices were subjectively identified in 850-hPa streamlines based on operational ECMWF analyses. All vortex positions refer to 00 UTC with the date given as numbers. Round symbols mark cyclonic systems (labelled I₁ and J), squares anticyclonic systems (labelled**
10 **I₂). Paler colours are used for days, when the vortices were not clearly identifiable. Vortices with a joint propagation are linked with dashed grey lines. The cores of significant 850-hPa jets are indicated with light blue arrows, again with the date at 00 UTC given as numbers. The boxes mark the areas used to compute the NSPD shown in Fig. 5a. The stippled lines shows the latitude range used to produce Fig. 9.**




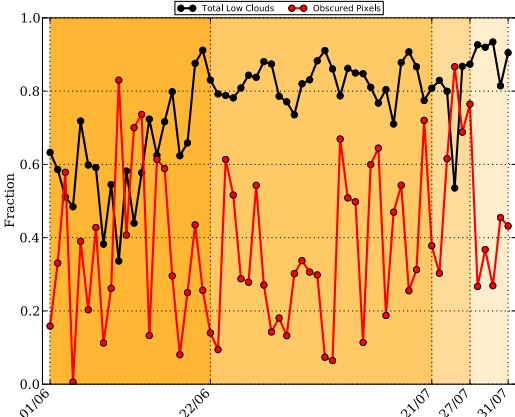

**Figure 17: Time evolution of low-cloud fraction over the DACCIWA focus region (5–10°N, 8°W–8°E, land pixels only, see Fig. 1 for location) during June-July 2016. Depicted as a black line is the average fraction of pixels covered with low clouds (i.e. cloud top pressure of 800 hPa or higher). Pixels that are covered with obscuring mid- or high-level clouds (fraction given as red line) are disregarded, leading to a relatively large uncertainty in the low-cloud estimates on some days. Basis for this analysis are daily 00 UTC CLAAS-2 images (see sect. 2.1). The four Phases of the DACCIWA campaign are marked with different orange shadings.**




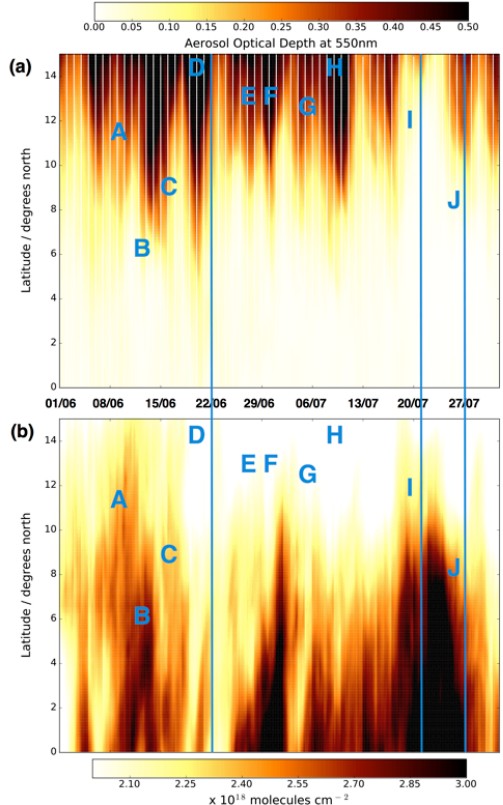

**Figure 18: Occurrence of mineral-dust (a) and CO (b) plumes over the DACCIWA region. Hovmöller plots for the longitude range 8°W–8E covering the whole June-July 2016 period generated from CAMS-IFS forecast data. Mineral dust plumes are indicated by DAOD at 550 nm and shows the transport of dust from north of the domain (i.e. Sahel/Sahara). The CO is column integrated and primarily shows the transport of biomass-burning influenced air from the SH. The four Phases of the DACCIWA campaign are indicated with blue lines. The most significant synoptic features A–J are marked at the approximate time and latitude of crossing the DACCIWA focus region.**





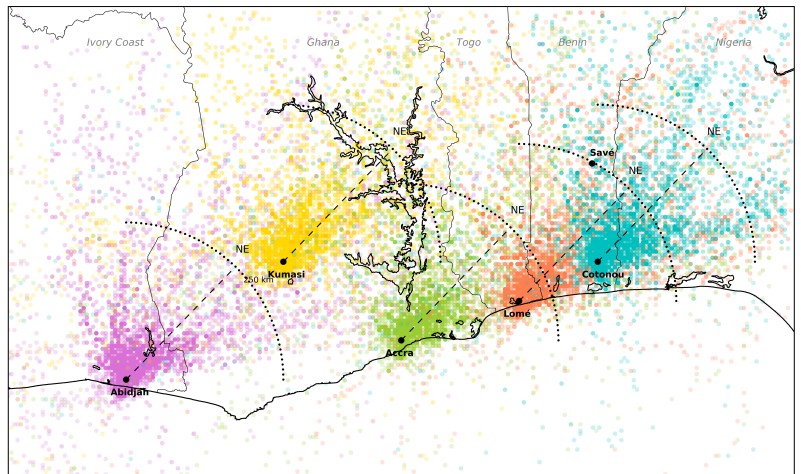

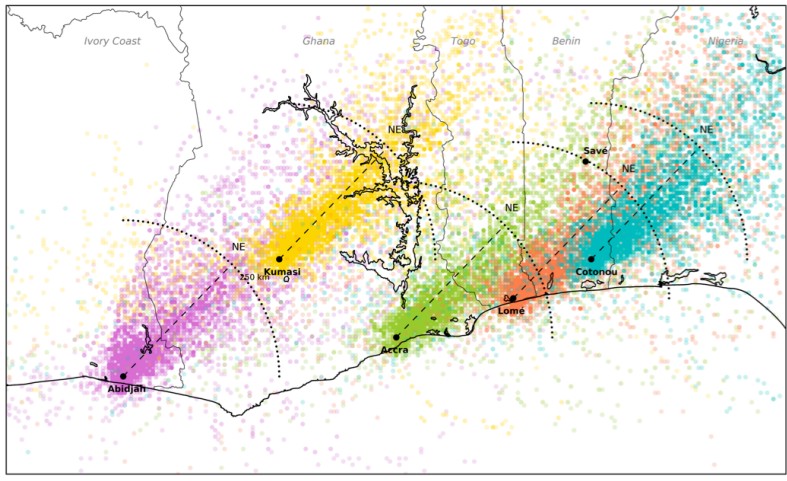

**Figure 19: Pollution plumes for the five main cities of interest during DACCIWA (Abidjan, Kumasi, Accra, Lomé, Cotonou)**
5 **during Phase 1 (top, 01–21 June 2016) and Phase 2 (22 June–20 July 2016). Shown are results from daily 24-hour simulations using the FLEXPART model (see sect. 2.2). Each dot is coloured according to its city source with the opacity increasing with the occurrences of pollution tracers at each point location. The quarter circles show distances of 250km around the main dispersion direction towards the northeast (dashed line).**





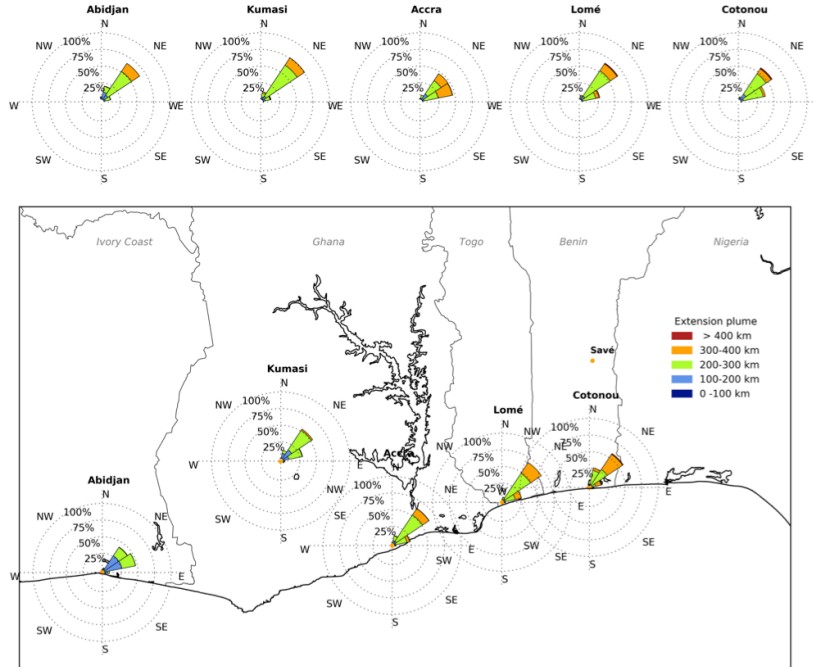

**Figure 20: Pollution roses of urban plumes from the five main cities of interest during DACCIWA (Abidjan, Kumasi, Accra, Lomé, Cotonou) during Phase 2 (22 June–20 July 2016). Each rose shows the predominant direction of plumes and their horizontal extension (colour code). The radius values correspond to the percentage of pollution plume direction occurrence in each**
5 **sector. Pollution roses derived from FLEXPART simulations are superimposed on the map. HYSPLIT derived pollution roses are at the top. See sect. 2.2 for more details.**