# Peer review of "A meteorological and chemical overview of the DACCIWA field campaign in West Africa in June–July 2016"

_Atmospheric Chemistry and Physics, 2017_

## Referee Comment (RC1) · S. JANICOT (Referee) · 5 Jun 2017

General comments : The aim of this manuscript is to provide the large-scale climatological context of the DACCIWA June-July 2016 field campaign and describe the behavior of the West African monsoon system into its specific phases, to characterize the most important synoptic scale weather systems affecting the Southern West Africa area and to discuss impacts on rainfall, clouds and atmospheric composition. A detailed description of the field activities will be provided in a companion paper (Flamant et al. 2017, submitted).

This type of work is essential for any field campaign in order to describe its general

context in terms of departures from the climatology and of occurrences of main weather system features within the campaign. The submitted manuscript is of a very high quality regarding these terms. It presents very clearly the up-to-date knowledge of the West African monsoon main features and uses them very accurately to highlight the different phases of the monsoon during June-July 2016, and then the main synoptic weather events within each of these phases. This is based on a rigorous analysis and use of the available data sets (not the new field measurements that will be analyzed in future papers) and up-to-date relevant tools. It is also quite synthetic and provides a very clear summary of the main features of the monsoon evolution and variability. So it will enable to investigate any field measurements in a well-known meteorological context and it let also space to investigate more any of the synoptic weather events identified here.

This manuscript addresses relevant scientific questions within the scope of ACP. It present novelties regarding the 2016 monsoon season and offers promising avenues to be addressed with the new DACCIWA measurements. Proper credit is given to previous work, in particular regarding the knowledge provided by the AMMA program and the available tools to detect and monitor synoptic weather systems in this area. The abstract provides a concise and complete summary. The manuscript is well structured and the results are very clearly presented.

Specific comments : A lot of diagnostics have been used in an accurate way to detect and monitor the main weather events during the DACCIWA campaign. I would like however to suggest testing some others to evaluate if they can provide additional clues in the interpretation of the monsoon evolution and variability in June-July 2016. I let it to the decision of the authors to see if some of them can be fruitful or not, because the manuscript is presently already in a relevant shape to be published :

- Regarding the SHL evolution, the diagnostics of Chauvin et al. (2010) on the quantification of east-west phases of SHL has been noticed and not used (the reference Roehrig et al., J. Climate, 2011, 5863-5878 should be indicated too). It defines an index different from Lavaysse et al's index and could provide complementary information and quantification.

- K. Cook (2015, J. Geophys. Res. Atmos., 120, 3085–3102, doi:10.1002/2014JD022579) introduced the inertial instability criterion to characterize the monsoon onset over West Africa. It could be useful to test it here for the monsoon onset of 2016.

- Poan et al. (2013, J. Atmosph. Sci., 70, 1035-1050) showed that the precipitable water variable can be a very useful indicator to monitor synoptic weather system trajectories (especially for westward AEW propagation over the Sahel). It can be evaluated on Hovmoeller diagrams as in Fig.9 and Fig.10.

- Regarding case studies H1-H2 and I1-I2 where jointly westward propagating coupled of cyclonic and anticyclonic vorticities have been detected, Diedhiou et al. (1999, Climate Dyn, 15, 795-822) identified such patterns during Gate Phase I and proposed a composite structure associated to a 6-9-day signal in atmospheric circulation associated with a clear modulation of the westerly/easterly wind component in the mid-levels as it is detected in 2016. More investigation in this way might provide additional information.

- The possible combination of AEW and MRG signals as it was presented by Cheng-Thorncroft-Kiladis, at the 2017 EGU session on Atmospheric composition, weather and climate in Sub-Saharan Africa might be interesting in future papers.

- Considering Fig.19, could you imagine one more figure as Fig.17 or Fig.18 to illustrate some variability of pollution plumes linked to the A-J weather events ? Or maps for very contrasted impact linked to two of these 10 events ?

Technical corrections : Use the widest possible space within the page for Fig.9 and Fig.10 because there is a lot of superimposed information that is not always clearly detectable for the reader.

The reference of the companion paper Flamant et al. (submitted) is missing in the reference list.

---

## Referee Comment (RC2) · T. Lefort (Referee) · 27 Jun 2017

General comments

This article describes both the large scale and subseasonal variability context, and the synoptic events that took place over a certain domain of west Africa. It is of very high relevance for future research papers. Moreover, the approach of confronting different time scales ( subseasonal versus synoptic) follows the seamless prediction advocated by WMO. Thus, it is also very useful for practicioners.

Concerning the synoptic scale, the paper shows that the conceptual models of

"Guinean systems" are maybe still to be investigated.

Specific comments

Between Sections 3 Large-scale settings, and Section 4 Detailed synoptic analysis, there is place for a section describing the intraseasonal variability. See Janicot et al, 2009, Large-scale overview of the summer monsoon over west Africa during the AMMA filed experiement in 2006; Section 3.4 Indeed, the different phases might have a direct link to the modes of variabilities.

Indeed, the MJO seems to play a role through propagative, favorable velocity potentiel at 200hPa for the phase 3. Since there are enough lines and colors on your figure 10, I suggest to add a Hovmöller diagram for MJO, Kelvin and ER only.

Technical corrections:

In Section 2.1 Data, you could maybe precise if the daily accumulation is from 00 to 00UTC or 06 to 06UTC, since comparing data from different sources might lead to discrepancies ( synops and ARC2 generally show 06-06UTC).
* * *

---

## Author Comment (AC1) · 26 Jul 2017

**Answers to Reviewers comments: Knippertz et al., A meteorological and chemical overview of the DACCIWA field campaign in West Africa in June–July 2016, ACP, doi.org/10.5194/acp-2017-345**

Please find our response in blue and the proposed changes in the manuscript in red.

**RC1 (Serge Janicot)**

*General comments : The aim of this manuscript is to provide the large-scale climatological context of the DACCIWA June-July 2016 field campaign and describe the behavior of the West African monsoon system into its specific phases, to characterize the most important synoptic scale weather systems affecting the Southern West Africa area and to discuss impacts on rainfall, clouds and atmospheric composition. A detailed description of the field activities will be provided in a companion paper (Flamant et al. 2017, submitted).*

*This type of work is essential for any field campaign in order to describe its general context in terms of departures from the climatology and of occurrences of main weather system features within the campaign. The submitted manuscript is of a very high quality regarding these terms. It presents very clearly the up-to-date knowledge of the West African monsoon main features and uses them very accurately to highlight the different phases of the monsoon during June-July 2016, and then the main synoptic weather events within each of these phases. This is based on a rigorous analysis and use of the available data sets (not the new field measurements that will be analyzed in future papers) and up-to-date relevant tools. It is also quite synthetic and provides a very clear summary of the main features of the monsoon evolution and variability. So it will enable to investigate any field measurements in a well-known meteorological context and it let also space to investigate more any of the synoptic weather events identified here.*

*This manuscript addresses relevant scientific questions within the scope of ACP. It present novelties regarding the 2016 monsoon season and offers promising avenues to be addressed with the new DACCIWA measurements. Proper credit is given to previous work, in particular regarding the knowledge provided by the AMMA program and the available tools to detect and monitor synoptic weather systems in this area. The abstract provides a concise and complete summary. The manuscript is well structured and the results are very clearly presented.*

Thank you very much for the time and effort spent on the review and your positive evaluation.

The following sentence will be added to the Acknowledgments: "The authors would like to thank Serge Janicot and Thierry Lefort for their effort to carefully review this paper and for their constructive criticism."

*Specific comments : A lot of diagnostics have been used in an accurate way to detect and monitor the main weather events during the DACCIWA campaign. I would like however to suggest testing some others to evaluate if they can provide additional clues in the interpretation of the monsoon evolution and variability in June-July 2016. I let it to the decision of the authors to see if some of them can be fruitful or not, because the manuscript is presently already in a relevant shape to be published :*

Thank you for the useful suggestions. We discuss them one by one below.

*- Regarding the SHL evolution, the diagnostics of Chauvin et al. (2010) on the quantification of east-west phases of SHL has been noticed and not used (the reference Roehrig et al., J. Climate, 2011, 5863-5878 should be indicated too). It defines an index different from*

*Lavaysse et al's index and could provide complementary information and quantification.*

The index developed by Chauvin et al. (2010) does not take into account the northward migration of the SHL, so it can detect an E or W phase even when the SHL is not located over the Sahara (before the onset or during extreme phases in summer when the SHL moves southward). This is a clear limitation when the period of study is close to the onset of the SHL. Moreover. The index reflects an E/W temperature anomaly that is possible to see in the longitude of the SHL location developed by Lavaysse et al. (2009), since it is based on the geopotentiel heights that are mainly controlled by the temperature. So the index of the E/W phases is a different method to detect the anomalies of temperature over the Sahara, but the results are quite closely related (at least for most significant anomalies) to those obtained with the method proposed by Lavaysse et al. (2009).

In section 2.2 we will add: "… which is closely linked to the East and West Phases of temperature anomalies proposed by Chauvin et al. (2010) when the SHL is located in its Saharan location (from end of June to mid-September)."

*- K. Cook (2015, J. Geophys. Res. Atmos., 120, 3085–3102, doi:10.1002/2014JD022579) introduced the inertial instability criterion to characterize the monsoon onset over West Africa. It could be useful to test it here for the monsoon onset of 2016.*

We calculated this index from 6-hourly ECMWF operational analysis. The result is shown below, demonstrating that the onset in 2016 was not characterised by significant inertial instability according to this measure. However, there is a marked drop when extratropical influences begin around 15 June and ahead of Phase 3.

[Figure]

*Fig. 1: Inertial stability index after Cook (2015) for the 01 June–31 July 2016 time period and the spatial window of 5° E to 5°W and 3°N to 6°N at 700hPa. Computations are based on 6-hourly ECMWF operational analysis data (red line). The blue line is a 3-day running mean. The shading shows the four Phases defined in the paper.*

We will add the following sentence to section 4.3: "This period is characterised by a substantial drop in the inertial stability index defined by Cook (2015) but negative values are only reached for short periods (not shown)."
We will add the following sentence to section 4.5: "…and low inertial stability according to the index defined by Cook (2015) (not shown)…"

*- Poan et al. (2013, J. Atmosph. Sci., 70, 1035-1050) showed that the precipitable water variable can be a very useful indicator to monitor synoptic weather system trajectories (especially for westward AEW propagation over the Sahel). It can be evaluated on Hovmoeller diagrams as in Fig.9 and Fig.10.*

As shown in our Fig. 7, column water vapour variations are rather small in the southern coastal region. As we are less interested in AEWs and more in the southern vortices, we decided not to use this diagnostic, which with no doubt is very useful for the Sahel.

No changes.

*- Regarding case studies H1-H2 and I1-I2 where jointly westward propagating coupled of cyclonic and anticyclonic vorticities have been detected, Diedhiou et al. (1999, Climate Dyn, 15, 795-822) identified such patterns during Gate Phase I and proposed a composite structure associated to a 6-9-day signal in atmospheric circulation associated with a clear modulation of the westerly/easterly wind component in the mid-levels as it is detected in 2016. More investigation in this way might provide additional information.*

Thank you for pointing this out to us. We checked the paper and carefully compared the results to our cases. For feature H a westward propagation speed of 7.1 m/s was estimated (see Table 1), similar to the 6–7 m/s from Diedhiou et al. The two centres of H are straddling the AEJ until the westerlies in between them become so strong that the automatic detection algorithm shifts the jet axis to the north of the cyclonic centre (see our Fig. 8). This is again qualitatively similar to the schematic Fig. 16c in Diedhiou et al. Feature I is much slower than H (4 m/s) and also connected with a northward shift of the AEJ to the north of the cyclonic centre. So this does not fit the 6–9-day wave regime so well. We decided to make the reader aware of this and added respective text to the introduction, results and conclusion section.

Reference will be added to the list.

End of 2$^{nd}$ paragraph of the Introduction, the following sentence will be added: "Diedhiou et al. (1999) present evidence for a more intermittent, slower (6–9 days period) wave regime with cyclonic and anticyclonic centres straddling the AEJ, longer wavelengths and an activity maximum over the continent in June and July."

Towards the end of section 4.4, the following additions will be made: "This propagating cyclonic-anticyclonic vortex couplet appears unrelated to any of the classical equatorial waves, but the slow propagation speed and the opposing circulation centres are consistent with the 6–9-day wave regime described by Diedhiou et al. (1999). To the best of our knowledge, the dynamical origin of such features is still somewhat unclear. In particular, the southern origin of the anticyclonic centre and its faster propagation seems unusual."

At the end of section 4.5 we will modify the text to read: "Mounier et al. (2008) also discuss enhanced westerly inflow, moist conditions and a Kelvin wave influence in connection with the QBZD but the match with their concept is hard to establish for a single case. Feature I is too slow to match the 6–9-day wave regime described by Diedhiou et al. (1999)."

Towards the end of the conclusion section we will add: "There are some similarities with the 6–9-day wave regime described by Diedhiou et al. (1999) but the dynamical causes are not clear."

*- The possible combination of AEW and MRG signals as it was presented by Cheng-Thorncroft-Kiladis, at the 2017 EGU session on Atmospheric composition, weather and climate in Sub-Saharan Africa might be interesting in future papers.*

We agree that this is interesting and relevant work, but as far as we can see, there is no publication about this idea yet. We decided to add a short sentence to the outlook at the end referring to http://meetingorganizer.copernicus.org/EGU2017/EGU2017-11491.pdf.

Reference will be added to the list.

We will add the following sentence at the end of the paper: "An interesting idea to be explored further in this context are interactions between AEWs and mixed Rossby-gravity waves as suggested by Cheng et al. (2017)."

*- Considering Fig.19, could you imagine one more figure as Fig.17 or Fig.18 to illustrate some variability of pollution plumes linked to the A-J weather events ? Or maps for very contrasted impact linked to two of these 10 events ?*

The problem with this is that the city pollution is most strongly affected by the wind in the lowest few hundred meters, where winds are quite stable throughout the pre- and post-onset periods (see for example Fig. 20). The synoptic features mostly modulate winds in the 850–700 hPa layer. This creates marked changes in dust and CO (Fig. 18) but not so much in the city plumes.

No changes.

*Technical corrections : Use the widest possible space within the page for Fig.9 and Fig.10 because there is a lot of superimposed information that is not always clearly detectable for the reader.*

Good point, we'll make it as wide as possible.

Figures 9 and 10 will be enlarged.

*The reference of the companion paper Flamant et al. (submitted) is missing in the reference list.*

The paper has just been accepted.

Reference will be added to list. "submitted" will be deleted from references in the text.

---

## Author Comment (AC2) · 26 Jul 2017

**Answers to Reviewers comments: Knippertz et al., A meteorological and chemical overview of the DACCIWA field campaign in West Africa in June–July 2016, ACP, doi.org/10.5194/acp-2017-345**

Please find our response in blue and the proposed changes in the manuscript in red.

**RC2 (Thierry Lefort)**

***General comments***

*This article describes both the large scale and subseasonal variability context, and the synoptic events that took place over a certain domain of west Africa. It is of very high relevance for future research papers. Moreover, the approach of confronting different time scales (subseasonal versus synoptic) follows the seamless prediction advocated by WMO. Thus, it is also very useful for practicioners. Concerning the synoptic scale, the paper shows that the conceptual models of "Guinean systems" are maybe still to be investigated.*

Thank you very much for the time and effort spent on the review and your positive evaluation.

The following sentence will be added to the Acknowledgments: "The authors would like to thank Serge Janicot and Thierry Lefort for their effort to carefully review this paper and for their constructive criticism."

***Specific comments***

*Between Sections 3 Large-scale settings, and Section 4 Detailed synoptic analysis, there is place for a section describing the intraseasonal variability. See Janicot et al, 2009, Large-scale overview of the summer monsoon over west Africa during the AMMA filed experiement in 2006; Section 3.4 Indeed, the different phases might have a direct link to the modes of variabilities.*

Janicot et al. (2008, not 2009) discuss the MJO, Kelvin waves and filtered OLR in the 10–25 days and 25–90 day windows. Kelvin waves, we do show and discuss already in connection with Fig. 10. According to our assessment, the MJO was not active during our period of interest and this is now mentioned in the text (see next answer). For other standard indices we checked the MISVA page (misva.sedoo.fr) and found some moderate activity in the SHL, Sahelian and QBZD indices. We now mention this in the text.

References to Janicot et al. (2011) and Roehrig et al. (2011) will be added.

In the Introduction we will add: "Other types of intraseasonal variability include the Sahelian and quasi-biweekly zonal dipole (QBZD) modes on timescales of 10–25 days (Janicot et al., 2011; Mounier et al., 2008; Roehrig et al., 2011)."

At the end of section 3, we will add: "This shows some resemblance with the SHL variations described by Chauvin et al. (2010) but on rather short timescales."

In section 4.1 we will add: "This modulation is consistent with the QBZD index showing a significant minimum around 14 June 2016 (see http://misva.sedoo.fr)." and then a little later in the same section "The moderate changes from drier to wetter and back to drier conditions in the Sahel during Phase 1 are reflected in weak but hardly significant undulations of the intraseaonal Sahelian index reaching a minimum on 12 June 2016 (see http://misva.sedoo.fr)."

In section 4.3 we will add: "According to http://misva.sedoo.fr, the intraseasonal SHL index reached a distinct maximum on 17 and 18 June 2016."

*Indeed, the MJO seems to play a role through propagative, favorable velocity potentiel at 200hPa for the phase 3. Since there are enough lines and colors on your figure 10, I suggest to add a Hovmöller diagram for MJO, Kelvin and ER only.*

We checked the classical RMM diagram (see figure below).

[Figure]

Based on this a widely used definition of MJO events is given here: http://www.cpc.ncep.noaa.gov/products/precip/CWlink/MJO/Composites/Precipitation/readme.shtml

According to this, no proper MJO event occurred during the period of interest. The signal in Phases 1 and 2 during July is too short-lived. We are going to mention this explicitly in the text and decided not to create an additional plot.

We will add a sentence to the end of the first paragraph of Section 3 saying: "Standard indices indicate that the MJO was not active over West Africa in June-July 2016 (not shown)."

*Technical corrections:*

*In Section 2.1 Data, you could maybe precise if the daily accumulation is from 00 to 00UTC or 06 to 06UTC, since comparing data from different sources might lead to discrepancies (synops and ARC2 generally show 06-06UTC).*

This uses TRMM times 00 UTC to 21 UTC every day, which corresponds to 2230–2230UTC. As we do not directly compare to other data sources, the exact timing is not crucial here, but we have added this information to the text anyway.

The respective part of Section 2.1 will read: "The temporal resolution of this product is 3 hourly, but here daily accumulations (2230–2230 UTC) are used for most investigations."